# Isoform-specific mutation in Dystonin-b gene causes late-onset protein aggregate myopathy and cardiomyopathy

**Nozomu Yoshioka[1,2], Masayuki Kurose[3], Masato Yano[1], Dang Minh Tran[1], Shujiro Okuda[4], Yukiko Mori-Ochiai[1], Masao Horie[5], Toshihiro Nagai[6], Ichizo Nishino[7], Shinsuke Shibata[6,8], Hirohide Takebayashi[1,9]***

[1]Division of Neurobiology and Anatomy, Graduate School of Medical and Dental Sciences, Niigata University, Niigata, Japan; [2]Transdisciplinary Research Programs, Niigata University, Niigata, Japan; [3]Department of Physiology, School of Dentistry, Iwate Medical University, Iwate, Japan; [4]Medical AI Center, School of Medicine, Niigata University, Niigata, Japan; [5]Department of Nursing, Niigata College of Nursing, Jōetsu, Japan; [6]Electron Microscope Laboratory, Keio University, Tokyo, Japan; [7]Department of Neuromuscular Research, National Institute of Neuroscience, National Center of Neurology and Psychiatry, Tokyo, Japan; [8]Division of Microscopic Anatomy, Graduate School of Medical and Dental Sciences, Niigata University, Niigata, Japan; [9]Center for Coordination of Research Facilities, Niigata University, Niigata, Japan

*For correspondence:
takebaya@med.niigata-u.ac.jp

**Abstract** Dystonin (*DST*), which encodes cytoskeletal linker proteins, expresses three tissue-selective isoforms: neural DST-a, muscular DST-b, and epithelial DST-e. *DST* mutations cause different disorders, including hereditary sensory and autonomic neuropathy 6 (HSAN-VI) and epidermolysis bullosa simplex; however, etiology of the muscle phenotype in *DST*-related diseases has been unclear. Because *DST-b* contains all of the *DST-a*-encoding exons, known HSAN-VI mutations could affect both DST-a and DST-b isoforms. To investigate the specific function of DST-b in striated muscles, we generated a *Dst-b*-specific mutant mouse model harboring a nonsense mutation. *Dst-b* mutant mice exhibited late-onset protein aggregate myopathy and cardiomyopathy without neuropathy. We observed desmin aggregation, focal myofibrillar dissolution, and mitochondrial accumulation in striated muscles, which are common characteristics of myofibrillar myopathy. We also found nuclear inclusions containing p62, ubiquitin, and SUMO proteins with nuclear envelope invaginations as a unique pathological hallmark in *Dst-b* mutation-induced cardiomyopathy. RNA-sequencing analysis revealed changes in expression of genes responsible for cardiovascular functions. In silico analysis identified *DST-b* alleles with nonsense mutations in populations worldwide, suggesting that some unidentified hereditary myopathy and cardiomyopathy are caused by *DST-b* mutations. Here, we demonstrate that the Dst-b isoform is essential for long-term maintenance of striated muscles.

## Editor's evaluation

The authors demonstrate that isoform-specific Dystonin-b (Dst-b) mutant mice show significant myopathy in skeletal and cardiac muscle at older ages without the peripheral neuropathy or post-natal lethality that are commonly observed by loss of function of the DST gene. The study provides novel information about the role of the Dst-b isoform in maintaining skeletal and cardiac muscle health. In addition, the study suggests that isoform-specific mutations in Dst-b gene may cause some hereditary skeletal and cardiac myopathies.

## Introduction

Skeletal and cardiac striated muscle fibers consist of a complex cytoskeletal architecture, and maintenance of the sarcomere structure is essential for muscle contraction. Dystonin (DST), also called bullous pemphigoid antigen 1 (BPAG1), is a cytoskeletal linker protein that belongs to the plakin family, which consists of DST, plectin, microtubule actin cross-linking factor 1 (MACF1), desmoplakin, and other plakins (*Boyer et al., 2010a*; *Künzli et al., 2016*; *Horie et al., 2017*). The *DST* gene consists of over 100 exons and generates tissue-selective protein isoforms through alternative splicing and different transcription initiation sites. At least three major DST isoforms exist, DST-a, DST-b, and DST-e, which are predominantly expressed in neural, muscular, and epidermal tissues, respectively (*Leung et al., 2001*). Although *DST-a* and *DST-b* share most of the same exons, *DST-b* contains five additional isoform-specific exons. Loss-of-function mutations in the *DST* locus have been reported to cause neurological disorders, hereditary sensory and autonomic neuropathy type 6 (HSAN-VI) (*Edvardson et al., 2012*), and the skin blistering disease epidermolysis bullosa simplex (*Groves et al., 2010*). DST-a is considered the crucial DST isoform in the pathogenesis of HSAN-VI because it is a neural isoform, and transgenic expression of Dst-a2 under the control of a neuronal promoter partially rescues disease phenotypes of *Dst^{Tg4}* homozygous mice, which is a mouse model of HSAN-VI (*Ferrier et al., 2014*). HSAN-VI patients exhibit muscular and cardiac abnormalities, such as reduced muscular action potential amplitude, muscle weakness, and disrupted cardiovascular reflexes (*Edvardson et al., 2012*; *Manganelli et al., 2017*; *Fortugno et al., 2019*; *Jin et al., 2020*; *Motley et al., 2020*). Because all reported HSAN-VI mutations could disrupt both *DST-a* and *DST-b*, it is unknown whether these muscle manifestations are caused by cell-autonomous effects of *DST-b* mutation and/or secondary effects of neurological abnormalities caused by *DST-a* mutation. Thus, the impact of *DST-b* deficiency in skeletal and cardiac muscles on the manifestations of HSAN-VI patients has not been fully elucidated.

*Dystonia musculorum* (*dt*) is a spontaneously occurring mutant in mice (*Duchen et al., 1964*; *Horie et al., 2016*) that results in sensory neuron degeneration, retarded body growth, dystonic and ataxic movements, and early postnatal lethality. The *Dst* gene was identified as a causative gene in *dt* mice (*Brown et al., 1995*; *Guo et al., 1995*), and later, *dt* mice were used as mouse models of HSAN-VI (*Ferrier et al., 2014*). *Dt* mice have been reported to display muscle weakness and skeletal muscle cytoarchitecture instability (*Dalpé et al., 1999*). We have also reported masseter muscle weakness in *Dst* gene trap (*Dst^{Gt}*) mice, neurodegeneration of the trigeminal motor nucleus, which innervates the masseter muscle, and muscle spindle atrophy in the masseter muscle (*Hossain et al., 2018*). Thus, known *Dst* mutant mice have mutations in both *Dst-a* and *Dst-b*, and exhibit abnormalities in neural and muscular tissues.

To investigate cell-autonomous functions of the Dst-b isoform in skeletal and cardiac muscles, we generated novel isoform-specific *Dst-b* mutant mice. *Dst-b* mutant mice displayed late-onset protein aggregate myopathy in skeletal and cardiac muscles without peripheral neuropathy and postnatal lethality. In this study, we first demonstrated the role of Dst-b in skeletal and cardiac muscle maintenance and that a *Dst-b* isoform-specific mutation causes protein aggregate myopathy, which has characteristics similar to myofibrillar myopathies (MFMs). We also observed conduction disorders in the electrocardiograms (ECGs) of *Dst-b* mutant mice. RNA sequencing (RNA-seq) analysis revealed changes in the expression of genes that are important for maintenance of cardiovascular structures and functions. These results suggest that the myopathic manifestations of patients with HSAN-VI may be caused by disruption of *DST-b*, in addition to neurological manifestations caused by *DST-a* mutation. Furthermore, because we identified a variety of *DST-b* mutant alleles with nonsense mutations worldwide using in silico analysis, it is possible that some unidentified hereditary myopathies and/or heart diseases are caused by isoform-specific mutations of the *DST-b* gene.

## Results

### Generation of novel isoform-specific *Dst-b* mutant mice

Among the three major Dst isoforms (*Figure 1A*), the *Dst-a* and *Dst-b* isoforms share the most exons, and the *Dst-b* isoform has five additional isoform-specific exons (*Figure 1B*). We generated a novel *Dst-b* mutant allele in which a nonsense mutation was introduced into a *Dst-b*-specific exon using the clustered regularly interspaced short palindromic repeats (CRISPR)/CRISPR-associated protein 9 (Cas9) genome editing method. The *Dst-b* mutant allele harbors a nonsense mutation followed by a

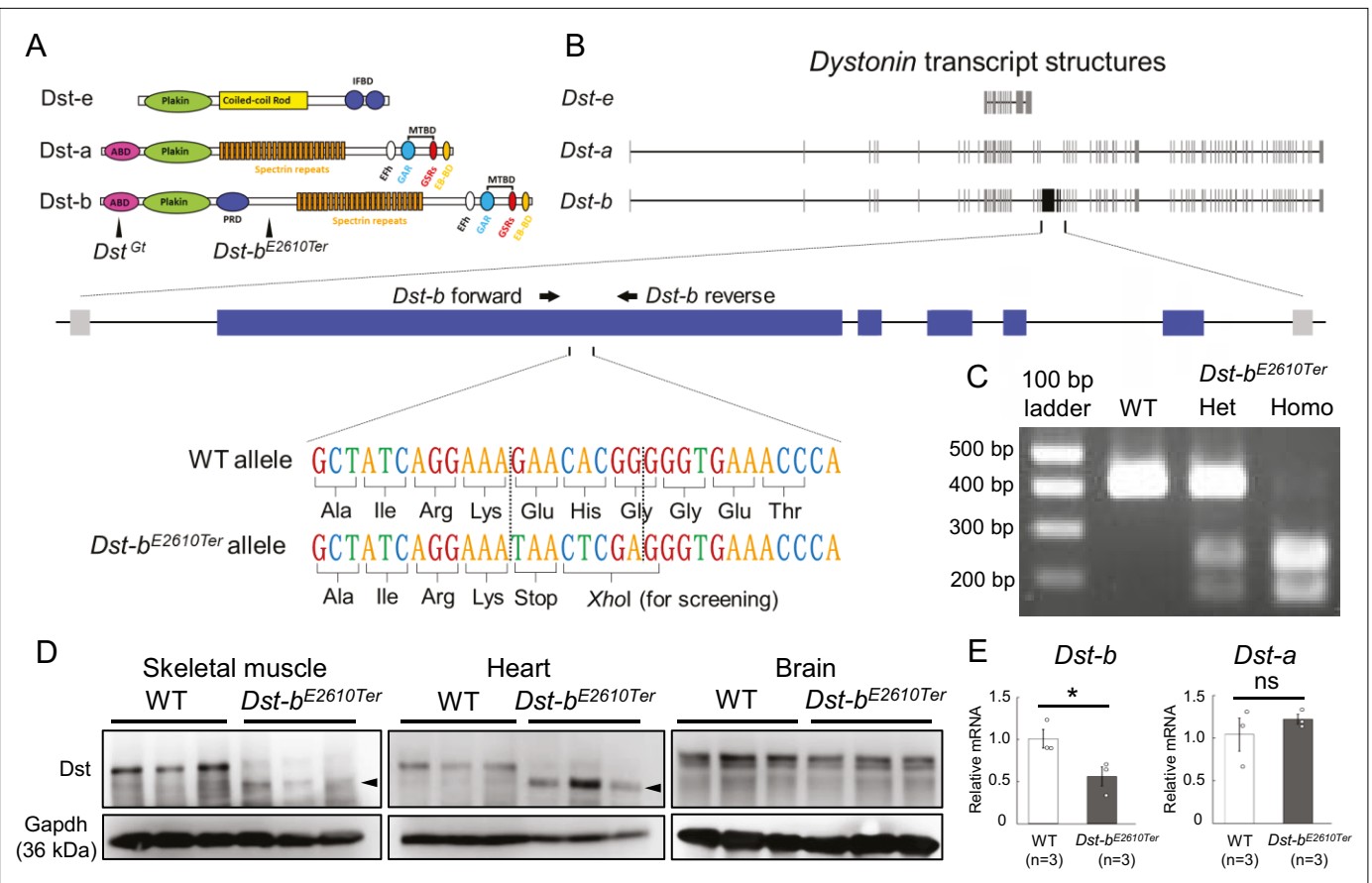

**Figure 1.** Generation of *Dst-b^E2610Ter* mutant mouse line. (**A**) The protein structure of Dst isoforms. The *Dst-b^E2610Ter* allele has the mutation between the plakin repeat domain (PRD) and the spectrin repeats. The *Dst^Gt* allele has the gene trap cassette within the actin-binding domain (ABD) shared by Dst-a and Dst-b isoforms. EB-BD, EB-binding domain; EFh, EF hand-calcium binding domains; GAR, growth arrest-specific protein 2-related domain; IFBD, intermediate filament-binding domain; MTBD, microtubule-binding domain. (**B**) A schematic representation of the *Dst* transcripts. The part of *Dst-b*-specific exons is enlarged, showing the mutation sites of the *Dst-b^E2610Ter* allele. The nonsense mutation and *Xho*I recognition sequence are inserted within the *Dst-b*-specific exon. The primer-annealing sites are indicated by arrows. (**C**) PCR-restriction fragment length polymorphism (RFLP) genotyping to distinguish WT, *Dst-b^E2610Ter* heterozygotes (Het), and *Dst-b^E2610Ter* homozygotes (Homo). PCR products from *Dst-b^E2610Ter* alleles are cut by *Xho*I. (**D**) Western blot analysis using the Dst antibody in lysates from the skeletal muscle (hindlimb muscle), heart, and brain (n = 3 mice, each genotype). Truncated Dst bands (arrowheads) were detected in the skeletal muscle and heart of *Dst-b^E2610Ter/E2610Ter* mice. Dst bands in the brain were unchanged between WT and *Dst-b^E2610Ter/E2610Ter*. Glyceraldehyde-3-phosphate dehydrogenase (Gapdh) was used as an internal control. (**E**) Quantitative PCR (qPCR) data of *Dst-a* and *Dst-b* mRNAs in the heart (n = 3 mice, each genotype). * denotes statistically significant difference at p<0.05 (*Dst-b*, p=0.0479) and ns means not statistically significant (*Dst-a*, p=0.4196), using Student's *t*-test. Data are presented as mean ± standard error (SE). Similar truncations of the Dst protein in three independent *Dst-b^E2610Ter* mouse lines are shown in *Figure 1—figure supplement 1*. The expression level of three N-terminal *Dst* isoforms is shown in *Figure 1—figure supplement 2*. qPCR data of *Dst* isoforms in the soleus are shown in *Figure 1—figure supplement 3*.

The online version of this article includes the following figure supplement(s) for figure 1:

**Figure supplement 1.** Expressions of truncated Dst-b from three *Dst-b^E2610Ter* alleles.

**Figure supplement 2.** Expressions for N-terminal isoforms of *Dst* in the cardiac tissue.

**Figure supplement 3.** Quantification of mRNA levels in the soleus.

*Xho*I site (*Figure 1B*). Restriction fragment length polymorphism (RFLP) analysis of PCR products was used to genotype mice (*Figure 1C*). Hereafter, this *Dst-b* mutant allele is referred to as the *Dst-b^E2610Ter* allele. In the heterozygous cross, approximately one-fourth of the progeny were *Dst-b* homozygous (*Dst-b^E2610Ter/E2610Ter*) mice, according to the Mendelian distribution. Western blotting was performed using an anti-Dst antibody that recognizes both Dst-a and Dst-b proteins. Dst-b bands were detected in tissue extracts from heart and skeletal muscle of wild-type (WT) mice, whereas shifted bands were observed in those of *Dst-b^E2610Ter* homozygous mice (*Figure 1D*). We analyzed three independent *Dst-b^E2610Ter* mouse lines, and similar truncations of the Dst protein were observed in the homozygotes

(*Figure 1—figure supplement 1*). In brain tissue extracts, Dst-a bands were equally detected between WT and *Dst-b^{E2610Ter/E2610Ter}* mice. Quantitative PCR (qPCR) analyses indicated that *Dst-b* mRNA was significantly reduced by approximately half in the cardiac tissues of *Dst-b^{E2610Ter/E2610Ter}* mice compared with those of WT mice, while *Dst-a* mRNA expression was unchanged (*Figure 1E*). Among three distinct isoforms of Dst-b (Dst-b1, Dst-b2, and Dst-b3) differing in their N-terminus (*Jefferson et al., 2006*), the expression level of *Dst* isoform1 mRNA was remarkably reduced in the cardiac tissue of *Dst-b^{E2610Ter/E2610Ter}* mice, whereas there was no significant change in the amounts of isoform 2 and isoform 3 mRNAs (*Figure 1—figure supplement 2A*). Similar changes in mRNA expression of *Dst* isoforms were observed in the soleus (*Figure 1—figure supplement 3A*). The frequency of exon usage was compared between cardiac and brain tissues (*Figure 1—figure supplement 2B*). *Dst-b*-specific exons were exclusively used in the heart and scarcely used in the brain. In the heart, the first exons of *Dst* isoform 1 and isoform 3 were more frequently used than those in the brain. However, *Dst* isoform 2 was predominantly used in the brain and weakly expressed in the heart. Thus, these findings suggest that *Dst* isoforms are differentially expressed among tissues and that the *Dst-b^{E2610Ter}* allele affects each *Dst* isoform to a different degree.

## Gross phenotypes of *Dst-b^{E2610Ter/E2610Ter}* mice

*Dst-b^{E2610Ter/E2610Ter}* mice had a normal appearance and seemed to have a normal life span. In the tail suspension test, 1-month-old *Dst-b^{E2610Ter/E2610Ter}* mice maintained normal postures, while gene trap mutant (*Dst^{Gt/Gt}*) mice with the *dt* phenotype displayed hindlimb clasping and dystonic movement (*Figure 2A*). Next, histological analysis was performed on the muscle spindles and dorsal root ganglia of 1-month-old mice, which are the main affected structures in *dt* mice. *Dst-b^{E2610Ter/E2610Ter}* mice exhibited normal muscle spindle structure, although the muscle spindles were markedly atrophied in *Dst^{Gt/Gt}* mice (*Figure 2B*). Similarly, neurofilament (NF) accumulation and induction of ATF3, a neural injury marker, were observed in the dorsal root ganglia of *Dst^{Gt/Gt}* mice but not in those of WT and *Dst-b^{E2610Ter/E2610Ter}* mice (*Figure 2C–E*). We also found that parvalbumin (*Pvalb*)-positive proprioceptive neurons were normally observed in *Dst-b^{E2610Ter/E2610Ter}* mice (*Figure 2F and G*) but were markedly decreased in *Dst^{Gt/Gt}* mice as previously described in other *dt* mouse lines (*Carlsten et al., 2001*). We next investigated the gross phenotypes of *Dst-b^{E2610Ter/E2610Ter}* mice over an extended time. *Dst-b^{E2610Ter/E2610Ter}* mice normally gained body weight until 1 year of age; then, they became lower body weight compared with WT mice after 1 year of age (*Figure 2H*). Some *Dst-b^{E2610Ter/E2610Ter}* mice exhibited kyphosis (data not shown). We have reported that *dt* mice display impairment of motor coordination as assessed by the rotarod test and wire hang test (*Horie et al., 2020*); however, WT and *Dst-b^{E2610Ter/E2610Ter}* mice older than 1 year exhibited normal motor coordination (*Figure 2I*). These data also support the idea that deficiency of the Dst-a isoform, but not the Dst-b isoform, is causative of *dt* phenotypes, including sensory neuron degeneration, abnormal movements, and postnatal lethality.

## Late-onset skeletal myopathy and cardiomyopathy in *Dst-b^{E2610Ter/E2610Ter}* mice

To further examine the skeletal muscle phenotype in aged *Dst-b^{E2610Ter/E2610Ter}* mice, we performed histological analyses. In the soleus muscle of *Dst-b^{E2610Ter/E2610Ter}* mice, small-caliber muscle fibers and centrally nucleated fibers (CNFs), which indicate muscle degeneration and regeneration, were frequently observed in mice at 16–23 months of age (*Figure 3A*) but were rare in 3–4-month-old *Dst-b^{E2610Ter/E2610Ter}* mice (*Figure 3—figure supplement 1A*). The percentage of CNFs was significantly increased in the soleus, gastrocnemius, and erector spinae muscles, with different severities (*Figure 3B*, *Figure 3—figure supplement 2*). Muscle mass of soleus normalized by body weight was not significantly different between control and *Dst-b^{E2610Ter/E2610Ter}* mice (control: 0.258 ± 0.012 [mg/g body weight] vs. *Dst-b^{E2610Ter/E2610Ter}*: 0.279 ± 0.015 [mg/g body weight]; control, n = 6 muscles from three mice; *Dst-b^{E2610Ter/E2610Ter}*, n = 6 muscles from three mice; p=0.3056, Student's *t*-test). Distribution of cross-sectional area in the soleus showed that small-caliber myofibers were abundant in *Dst-b^{E2610Ter/E2610Ter}* mice compared with WT mice (*Figure 3C*). Masson's trichrome staining indicated skeletal muscle fibrosis and, particularly, a marked increase in the connective tissue surrounding small-caliber muscle fibers (*Figure 3D and E*) in the soleus muscle of *Dst-b^{E2610Ter/E2610Ter}* mice. Cardiac fibrosis was also observed in *Dst-b^{E2610Ter/E2610Ter}* mouse hearts (*Figure 3D and E*) but was not observed in the hearts of 3–4-month-old *Dst-b^{E2610Ter/E2610Ter}* mice (*Figure 3—figure supplement 1B*).

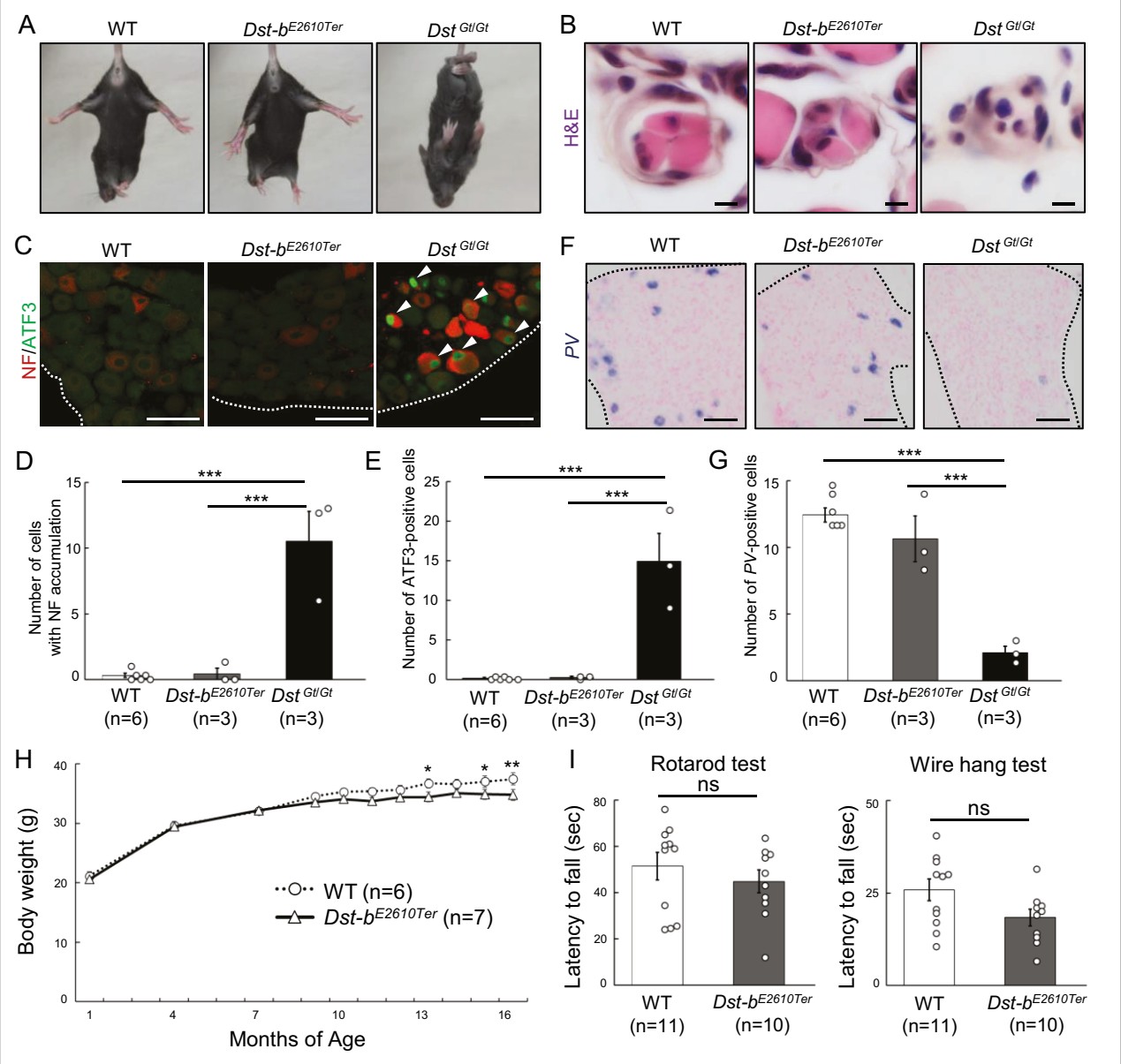

**Figure 2.** Characterizations of gross phenotypes of *Dst-b*$^{E2610Ter}$ homozygous mice. (**A**) Tail suspension test of WT, *Dst-b*$^{E2610Ter/E2610Ter}$, and *Dst*$^{Gt/Gt}$ mice around 1 month old. WT and *Dst-b*$^{E2610Ter/E2610Ter}$ mice exhibited normal posture, whereas the *Dst*$^{Gt/Gt}$ mice showed hindlimb clasping. (**B**) Muscle spindle structure on the cross sections of soleus with H&E staining. Muscle spindles of *Dst-b*$^{E2610Ter/E2610Ter}$ mice appeared normal, while *Dst*$^{Gt/Gt}$ mice showed atrophy of the intrafusal muscle fiber in muscle spindles around 1 month old. (**C**) Double immunohistochemistry (IHC) of neurofilament (NF) and ATF3 on the sections of dorsal root ganglia (DRG) around 1 month old. In only *Dst*$^{Gt/Gt}$ mice, NF was accumulated in some DRG neurons, and the neural injury marker, ATF3 was expressed. (**D, E**) Quantitative data of numbers of NF-accumulating cells (**D**) and ATF3-positive cells (**E**) in DRG of WT, *Dst-b*$^{E2610Ter/E2610Ter}$, and *Dst*$^{Gt/Gt}$ mice (n = 6 WT mice; n = 3 *Dst-b*$^{E2610Ter/E2610Ter}$ mice; n = 3 *Dst*$^{Gt/Gt}$ mice). *** denotes statistically significant difference at p<0.005 (NF, WT vs. *Dst*$^{Gt/Gt}$, p=0.0000; *Dst-b*$^{E2610Ter/E2610Ter}$ vs. *Dst*$^{Gt/Gt}$, p=0.0001; ATF3, WT vs. *Dst*$^{Gt/Gt}$, p=0.0001; *Dst-b*$^{E2610Ter/E2610Ter}$ vs. *Dst*$^{Gt/Gt}$, p=0.0002), using ANOVA. (**F**) Parvalbumin (*PV*) ISH on the section of DRG around 1 month old. *PV*-positive proprioceptive neurons were greatly decreased in DRG of *Dst-b*$^{E2610Ter/E2610Ter}$ mice but not in that of *Dst-b*$^{E2610Ter/E2610Ter}$ mice. Dotted lines indicate the edge of DRG. (**G**) Quantitative data of number of *PV*-positive cells in DRG of WT, *Dst-b*$^{E2610Ter/E2610Ter}$, and *Dst*$^{Gt/Gt}$ mice (n = 6 WT mice; n = 3 *Dst-b*$^{E2610Ter/E2610Ter}$ mice; n = 3 *Dst*$^{Gt/Gt}$ mice). *** denotes statistically significant difference at p<0.005 (*PV*, WT vs. *Dst*$^{Gt/Gt}$, p=0.0000; *Dst-b*$^{E2610Ter/E2610Ter}$ vs. *Dst*$^{Gt/Gt}$, p=0.0002), using ANOVA. (**H**) Male *Dst-b*$^{E2610Ter/E2610Ter}$ mice normally gained body weight until 1 year old and then became lighter than male WT mice (n = 6 WT mice; n = 7 *Dst-b*$^{E2610Ter/E2610Ter}$ mice; two-way ANOVA, genotype effect: p=0.1213; age effect: p=0.0000; genotype × age interaction: p=0.0398). * and ** denote statistically significant difference at p<0.05 (13 months old, p=0.0197; 15 months old, p=0.0296) and p<0.01 (16 months old, p=0.0073). (**I**) Behavior tests to assess motor coordination and grip strength around 1 year old. *Dst-b*$^{E2610Ter/E2610Ter}$ mice showed normal motor coordination and grid strength (n = 11 WT mice; n = 10 *Dst-b*$^{E2610Ter/E2610Ter}$ mice). ns means not statistically significant (rotarod test, p=0.4126; fire hang test, p=0.0612), using Student's *t*-test. Data are presented as mean ± standard error (SE). Scale bars: (**B**) 5 µm; (**C, D**) 50 µm.

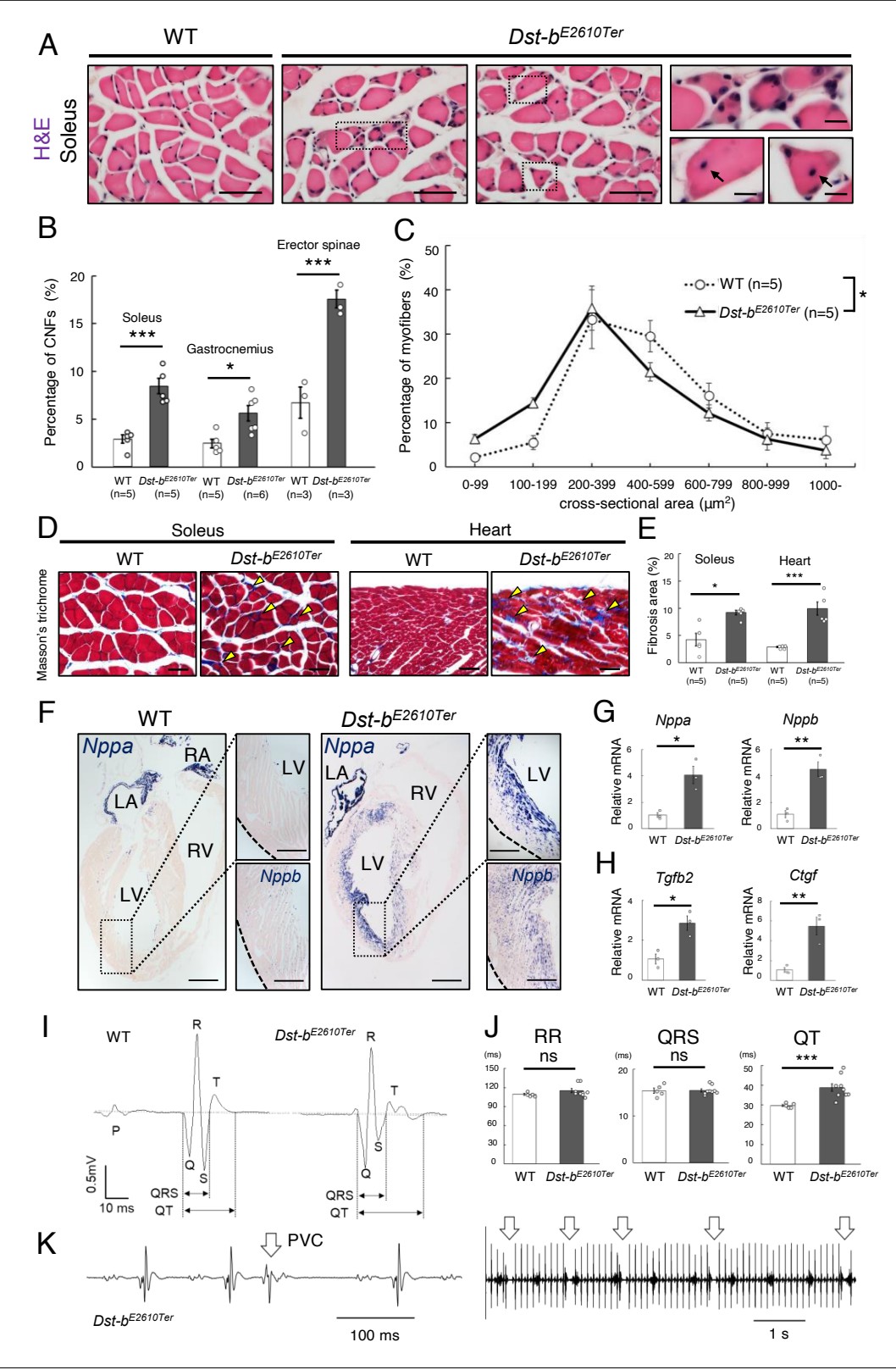

**Figure 3.** Pathological alterations in skeletal and cardiac muscles of *Dst-b*[E2610Ter] mice. (**A**) H&E-stained cross soleus sections of WT and *Dst-b*[E2610Ter/E2610Ter] mice at 23 months of age. Dotted boxes in the soleus of *Dst-b*[E2610Ter/E2610Ter] include small-caliber and centrally nucleated fibers (CNFs, arrows) shown as insets. Histological analysis of soleus of *Dst-b*[E2610Ter/E2610Ter] mice at 3–4 months of age are shown in **Figure 3—figure supplement 1**. Histological

*Figure 3 continued on next page*

*Figure 3 continued*

analysis of gastrocnemius and erector spinae is shown in *Figure 3—figure supplement 2*. (**B**) Quantitative data representing percentages of CNFs in the soleus (n = 5 mice, each genotype), gastrocnemius (n = 5 WT mice; n = 6 *Dst-b*$^{E2610Ter/E2610Ter}$ mice), and erector spinae (n = 3 WT mice; n = 3 *Dst-b*$^{E2610Ter/E2610Ter}$ mice). * and *** denote statistically significant difference at p<0.05 (gastrocnemius, p=0.0118), and p<0.005 (soleus, p=0.0009; erector spinae, p=0.0044), using Student's *t*-test. (**C**) Distribution of cross-sectional area in the soleus of WT and *Dst-b*$^{E2610Ter/E2610Ter}$ mice (n = 5 mice, each genotype, at 16–23 months old; two-way ANOVA; genotype effect: p=0.0133; area effect: p=0.0000; genotype × area interaction: p=0.2032). (**D**) Masson's trichrome staining showed a fibrosis in the soleus and heart of *Dst-b*$^{E2610Ter/E2610Ter}$ mice at 16 months of age (arrowheads). (**E**) Quantitative data of the extent of fibrosis in soleus and heart of WT and *Dst-b*$^{E2610Ter/E2610Ter}$ mice (n = 5 mice, each genotype, at 16–23 months old). * and *** denote statistically significant difference at p<0.05 (p=0.0109) and p<0.005 (p=0.0045), using Student's *t*-test. (**F**) *Nppa* and *Nppb* ISH in the heat at 16 months of age. *Nppa* mRNA was upregulated in the myocardium of left ventricle (LV) of *Dst-b*$^{E2610Ter/E2610Ter}$ mice, but not in the right ventricle (RV). Dotted areas are shown as insets. *Nppa* mRNA was strongly expressed in the myocardium of left atrium (LA) and right atrium (RA). Insets below represents *Nppb* mRNA the in same regions. (**G, H**) qPCR analyses on cardiac stress markers (**G**) and profibrotic cytokines (**H**) (n = 3 mice, each genotype, at 20–25 months old). * and ** denote statistically significant difference at p<0.05 (*Nppa*, p=0.0119; *Tgfb2*, p=0.0149) and p<0.01 (*Nppb*, p=0.0053; *Ctgf*, p=0.0095), using Student's *t*-test. (**I**) Representative electrocardiogram (ECG) images of P-QRS-T complex. (**J**) Intervals of RR, QRS, and QT were quantified (n = 5 WT mice; n = 9 *Dst-b*$^{E2610Ter/E2610Ter}$ mice, at 18–25 months old). *** denotes statistically significant difference at p<0.005 (QT, p=0.0044) and ns means not statistically significant (RR, p=0.2000; QRS, p=0.8964), using Student's *t*-test. (**K**) Premature ventricular contractions (PVC, arrows) were recorded in short-range (left panel) and long-range (right panel) ECG images from a *Dst-b*$^{E2610Ter/E2610Ter}$ mouse. Data are presented as mean ± SE. Scale bars: (**A** (left images), **D**) 50μm; (**A** (right magnified images)) 5 μm; (**F**) 1mm; (**F** (magnified images)) 200 μm.

The online version of this article includes the following figure supplement(s) for figure 3:

**Figure supplement 1.** Histological analysis in the soleus and heart of young *Dst-b*$^{E2610Ter}$ mice.

**Figure supplement 2.** Histological analysis in gastrocnemius and erector spinae of *Dst-b*$^{E2610Ter}$ mice.

To assess the influence of the *Dst-b* mutation on cardiomyocytes, expression of the heart failure markers natriuretic peptide A (*Nppa*) and natriuretic peptide B (*Nppb*) (*Sergeeva and Christoffels, 2013*) was investigated. *Nppa* mRNA was markedly upregulated in the myocardium of the left ventricle in 16-month-old *Dst-b*$^{E2610Ter/E2610Ter}$ mice and was not increased in that of WT mice (*Figure 3F*). *Nppb* mRNA was also upregulated in the left ventricle myocardium in *Dst-b*$^{E2610Ter/E2610Ter}$ mice (*Figure 3F*). qPCR analysis also demonstrated increased expression of *Nppa* and *Nppb* mRNAs in the cardiac tissue of *Dst-b*$^{E2610Ter/E2610Ter}$ mice (*Figure 3G*). Expression of profibrotic cytokines, including transforming growth factor beta-2 (*Tgfb2*) and connective tissue growth factor (*Ctgf*) mRNAs, was also remarkably increased in *Dst-b*$^{E2610Ter/E2610Ter}$ mice (*Figure 3H*). At 3–4 months of age, the *Nppa* transcript was slightly upregulated in *Dst-b*$^{E2610Ter/E2610Ter}$ mouse hearts without obvious cardiac fibrosis (*Figure 3—figure supplement 1B*). To assess the cardiac function, ECGs were recorded from anesthetized WT and *Dst-b*$^{E2610Ter/E2610Ter}$ mice at 18–25 months of age (*Figure 3I*). *Dst-b*$^{E2610Ter/E2610Ter}$ mice exhibited remarkably prolonged QT intervals compared with those of WT mice (*Figure 3J*), which suggests abnormal myocardial repolarization. However, the RR interval and QRS duration were not different between WT and *Dst-b*$^{E2610Ter/E2610Ter}$ mice (*Figure 3J*). In addition, premature ventricular contractions (PVCs) were recorded in two of nine *Dst-b*$^{E2610Ter/E2610Ter}$ mice analyzed (*Figure 3K*). Individual *Dst-b*$^{E2610Ter/E2610Ter}$ mice harboring PVCs showed severe cardiac fibrosis and remarkable upregulation of *Nppa* mRNA in the ventricular myocardium (data not shown). Such ECG abnormalities were not observed in 3-month-old *Dst-b*$^{E2610Ter/E2610Ter}$ mice (RR of WT mice: 134.2 ± 26.6 ms vs. *Dst-b*$^{E2610Ter/E2610Ter}$ mice: 138.8 ± 33.5 ms; p=0.8234, Student's *t*-test; QRS of WT mice: 17.3 ± 2.0 ms vs. *Dst-b*$^{E2610Ter/E2610Ter}$ mice: 17.4 ± 1.2 ms; p=0.9514, Student's *t*-test; QT of WT mice: 31.7 ± 4.3 ms vs. *Dst-b*$^{E2610Ter/E2610Ter}$ mice: 31.2 ± 2.3 ms; p=0.8599, Student's *t*-test; WT, n = 4 mice; *Dst-b*$^{E2610Ter/E2610Ter}$, n = 5 mice). These data indicate that the *Dst-b* isoform-specific mutation causes late-onset cardiomyopathy and cardiac conduction disturbance.

## Protein aggregations in myofibers of *Dst-b*$^{E2610Ter/E2610Ter}$ mice

MFM is a type of hereditary myopathy that is defined on the basis of common pathological features, such as myofibril disorganization beginning at the Z-disks and ectopic protein aggregation in myocytes (*De Bleecker et al., 1996*; *Nakano et al., 1996*). Desmin is an intermediate filament protein located in

Z-disks, and aggregation of desmin and other cytoplasmic proteins is a pathological hallmark of MFM (*De Bleecker et al., 1996*; *Clemen et al., 2013*; *Batonnet-Pichon et al., 2017*). First, we performed desmin immunohistochemistry (IHC) and found abnormal desmin aggregation in the skeletal muscle of *Dst-b$^{E2610Ter/E2610Ter}$* mice at 16–23 months of age (*Figure 4A and B*). In the soleus of *Dst-b$^{E2610Ter/E2610Ter}$* mice, desmin massively accumulated in the subsarcolemmal region, whereas in that of WT mice, desmin protein was predominantly located underneath the sarcolemma (*Figure 4A*). Desmin aggregates were also observed in the cardiomyocytes of *Dst-b$^{E2610Ter/E2610Ter}$* mice older than 16 months (*Figure 4—figure supplement 1*). In longitudinal sections of *Dst-b$^{E2610Ter/E2610Ter}$* soleus muscle, localization of desmin to the Z-disks was mostly preserved; however, displacement of Z-disks was occasionally observed in the muscle fibers bearing subsarcolemmal desmin aggregates (*Figure 4C*). At 3–4 months of age, desmin aggregates were scarcely observed in *Dst-b$^{E2610Ter/E2610Ter}$* mouse soleus (*Figure 3—figure supplement 1A*).

Next, we investigated the molecular composition of the protein aggregates in *Dst-b$^{E2610Ter/E2610Ter}$* mouse muscle fibers. We found that αB-crystallin was co-aggregated with desmin in the subsarcolemmal regions of the *Dst-b$^{E2610Ter/E2610Ter}$* soleus (*Figure 4D*). αB-crystallin is a small heat shock protein that binds to desmin and actin and is known to accumulate in muscles of patients with MFMs (*Clemen et al., 2013*). In addition, plectin had also accumulated in desmin-positive aggregates of the *Dst-b$^{E2610Ter/E2610Ter}$* soleus (*Figure 4E*). Plectin is a cytoskeletal linker protein that is normally located in the Z-disks. Because protein aggregates in muscles affected by MFM often contain the Z-disk protein encoded by the causative gene itself, we examined subcellular distribution of myotilin that is a Z-disk component encoded by a causative gene for MFM. We found that myotilin was accumulated in some myofibers of the *Dst-b$^{E2610Ter/E2610Ter}$* soleus (*Figure 4—figure supplement 2*). Furthermore, we evaluated the subcellular distribution of truncated Dst proteins in the *Dst-b$^{E2610Ter/E2610Ter}$* soleus. As expected, truncated Dst proteins accumulated in subsarcolemmal aggregates of desmin in the *Dst-b$^{E2610Ter/E2610Ter}$* soleus (*Figure 4F*). In addition, some small-caliber myofibers exhibited cytoplasmic accumulation of truncated Dst proteins surrounding nuclei. Dst proteins were also distributed on Z-disks (*Figure 4G*), as reported in previous studies (*Boyer et al., 2010a*; *Steiner-Champliaud et al., 2010*). Truncated Dst proteins were also localized in normally shaped Z-disks in the *Dst-b$^{E2610Ter/E2610Ter}$* soleus and were dispersed from disorganized Z-disks (*Figure 4G*). Taken together, these data indicate that protein aggregates in *Dst-b$^{E2610Ter/E2610Ter}$* mouse muscles are composed of a variety of proteins, similar to aggregates found in muscles affected by other MFMs.

## Mitochondrial structure and gene expression alterations in striated muscles of *Dst-b$^{E2610Ter}$* mice

Mitochondrial abnormalities have also been reported in muscles of patients with MFMs and animal models (*Joshi et al., 2014*; *Bührdel et al., 2015*; *Winter et al., 2016*). We investigated the mitochondrial distribution using cytochrome C and Tom20 IHC and found abnormal accumulation of mitochondria underneath the sarcolemma in some fibers of *Dst-b$^{E2610Ter/E2610Ter}$* soleus muscles (*Figure 5A*). Double staining with cytochrome C and desmin revealed that mitochondrial accumulation occurred in the same region as desmin aggregates (*Figure 5C*). We also found a strong phosphorylated PERK signal, an organelle stress sensor, in the subsarcolemmal spaces with accumulated mitochondria (*Figure 5D*).

Next, we quantified the mRNA expression levels of several genes responsible for oxidative phosphorylation (*Ndufb8*, *Sdha*, *Uqcrc2*, *Cox4i1*, and *Atp5a1*) in the soleus (*Figure 5E*). The expression of several genes, including *Ndufb8*, *Sdha*, and *Cox4i1*, was significantly reduced in *Dst-b$^{E2610Ter/E2610Ter}$* mice, suggesting mitochondrial dysfunction in the skeletal muscle of *Dst-b$^{E2610Ter/E2610Ter}$* mice. Changes in genes responsible for oxidative phosphorylation suggest the possibility of muscle fiber type switch and/or oxidative stress (*Bonnard et al., 2008*; *Zhang et al., 2017*). However, quantification of expression levels of muscle fiber-type marker genes and oxidative stress-related genes showed no changes in the soleus of *Dst-b$^{E2610Ter/E2610Ter}$* mice (*Figure 1—figure supplement 3B and C*). At 3–4 months of age, mitochondrial accumulation was already observed in soleus of *Dst-b$^{E2610Ter/E2610Ter}$* mice before the appearance of CNF and desmin aggregates (*Figure 3—figure supplement 1A*). Therefore, mitochondrial abnormalities may precede protein aggregation and muscle degeneration.

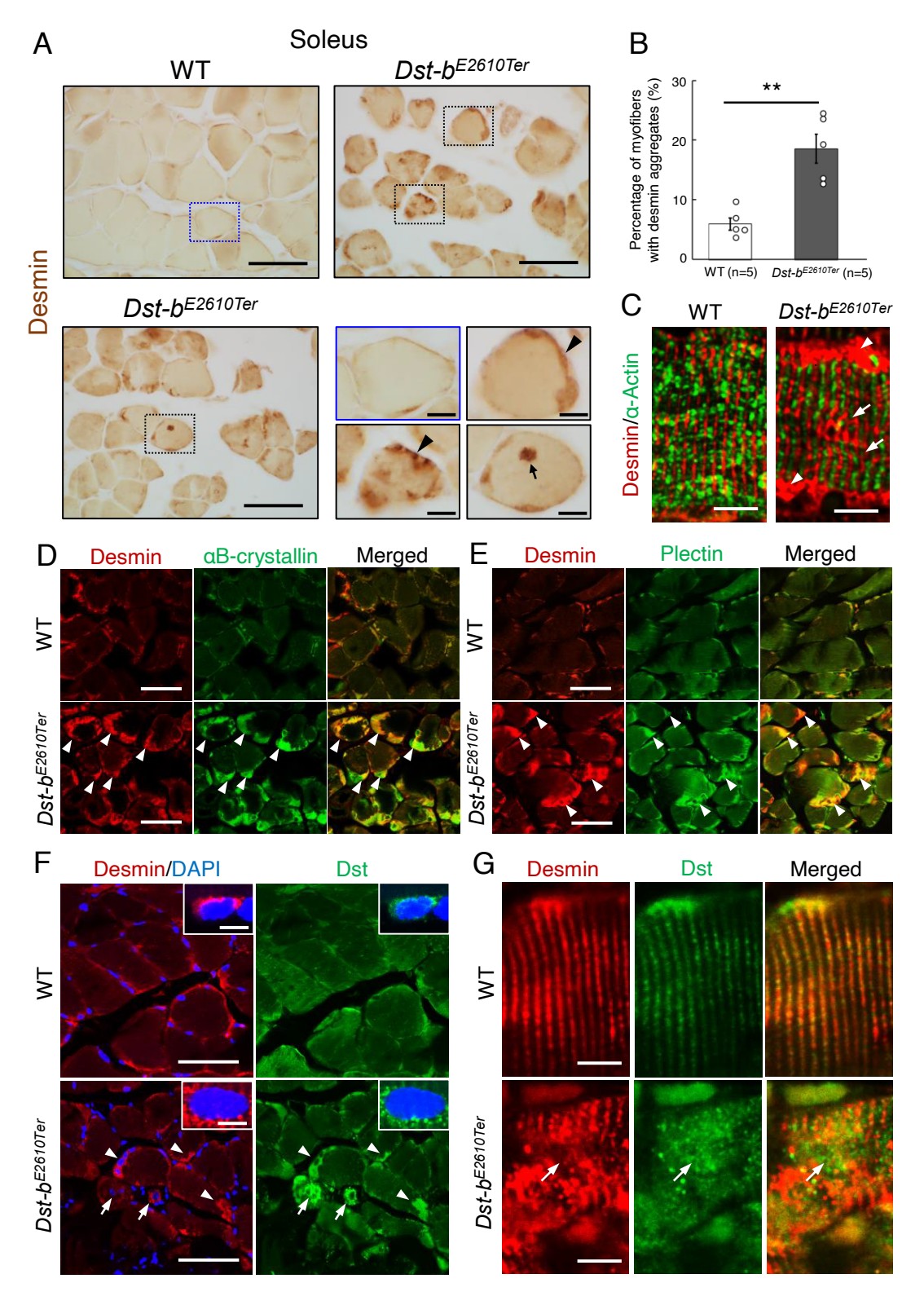

**Figure 4.** *Dst-b*[E2610Ter] mutation leads to protein aggregation myopathy. (**A**) Desmin immunohistochemistry (IHC) on the cross sections of soleus from WT and *Dst-b*[E2610Ter/E2610Ter] mice at 23 months of age using anti-desmin antibody (RD301). Dotted boxes in *Dst-b*[E2610Ter/E2610Ter] soleus indicate desmin aggregates underneath the sarcolemma (arrowheads) and in the sarcoplasmic region (arrow) shown as insets. Desmin IHC on heart of *Dst-b*[E2610Ter/E2610Ter] mice is shown in *Figure 4—figure supplement 1*. (**B**) Quantitative data of the percentage of myofibers with desmin aggregates in soleus of WT and

*Figure 4 continued*

*Dst-b^E2610Ter/E2610Ter* mice (n = 5 mice, each genotype, at 16–23 months old). \*\* denotes statistically significant difference at p<0.01 (p=0.0051), using Student's *t*-test. Data are presented as mean ± SE. (**C**) Immunofluorescent images of longitudinal soleus sections labeled with anti-desmin (rabbit IgG) and anti-alpha-actin antibodies. Muscle fibers harboring subsarcolemmal desmin aggregates (arrowheads) showed the Z-disk displacement (arrows). (**D, E**) Double IHC using anti-desmin (RD301) and anti-αB-crystallin antibodies (**D**) or anti-desmin (Rabbit IgG) and anti-plectin antibodies (**E**) on the cross sections of soleus. In *Dst-b^E2610Ter/E2610Ter* soleus, αB-crystallin and plectin were accumulated in subsarcolemmal regions with desmin (white arrowheads). Images of myotilin, a Z-disk component, are shown in *Figure 4—figure supplement 2*. (**F**) Double IHC using anti-desmin antibody and anti-Dst antibody on cross sections of WT and *Dst-b^E2610Ter/E2610Ter* soleus. Dst protein accumulated in subsarcolemmal desmin aggregates (white arrowheads). Dst protein was also accumulated around myonuclei-labeled with DAPI (arrows). Insets show localizations of desmin and Dst proteins around nuclei. (**G**) Double IHC of desmin and Dst. In longitudinal soleus sections, Dst protein was distributed in a striped pattern at desmin-positive Z-disks. In *Dst-b^E2610Ter/E2610Ter* soleus sections, Dst protein was dispersed around displaced Z-disks (arrows). Scale bars: (**A, D, E, F**) 50μm; (**A** (magnified images), **C, F** (insets), **G**) 5μm.

The online version of this article includes the following figure supplement(s) for figure 4:

**Figure supplement 1.** Desmin immunohistochemistry (IHC) on the sections of heart from WT and *Dst-b^E2610Ter/E2610Ter* mice at 23 months of age using anti-desmin antibody (RD301).

**Figure supplement 2.** Myotilin immunohistochemistry (IHC) on the cross sections of soleus from WT and *Dst-b^E2610Ter/E2610Ter* mice.

## Ultrastructural changes in skeletal and cardiac muscles of *Dst-b^E2610Ter/E2610Ter* mice

Transmission electron microscope (TEM) analysis was performed to further investigate the ultrastructural features of skeletal and cardiac muscles of *Dst-b^E2610Ter/E2610Ter* mice. In the *Dst-b^E2610Ter/E2610Ter* soleus, focal dissolution of myofibrils and Z-disk streaming were observed at 22 months of age (*Figure 6A*). Subsarcolemmal regions of the *Dst-b^E2610Ter/E2610Ter* soleus were often filled with accumulated mitochondria and granulofilamentous materials including focal accumulation of glycogen granules, electron-dense disorganized Z-disks, and electron-pale filamentous materials (*Figure 6B*). These observations have been reported as typical features of MFMs (*Batonnet-Pichon et al., 2017*). Furthermore, we found dysmorphic nuclei in the soleus of *Dst-b^E2610Ter/E2610Ter* mice (*Figure 6C*).

Focal myofibrillar dissolution and mitochondrial accumulation were observed in *Dst-b^E2610Ter/E2610Ter* mouse cardiomyocytes (*Figure 6D*). We often observed an invaginated nuclear envelope and a structure similar to the nucleoplasmic reticulum (*Bezin et al., 2008*; *Malhas et al., 2011*) with cytoplasmic components in the nucleus (*Figure 6E*). Interestingly, inclusions containing crystalline structures were frequently observed in dysmorphic nuclei, which were a unique feature of *Dst-b^E2610Ter/E2610Ter* cardiomyocytes. Only a limited number of reports have described myopathy with intranuclear inclusions (*Oteruelo, 1976*; *Oyer et al., 1991*; *Weeks et al., 2003*; *Ogasawara et al., 2020*).

## Abnormal nuclear structures in *Dst-b^E2610Ter* cardiomyocytes

The observation of unique nuclear inclusions in dysmorphic nuclei in heart tissue prompted us to perform detailed histological analyses of *Dst-b^E2610Ter/E2610Ter* mouse cardiomyocytes. Upon H&E staining, eosinophilic structures were observed in *Dst-b^E2610Ter/E2610Ter* cardiomyocyte nuclei (*Figure 7A*). Dysmorphic nuclei were clearly observed by lamin A/C IHC, showing a deeply invaginated nuclear envelope devoid of DNA (*Figure 7B*). In TEM observations, nuclear crystalline inclusions and cytoplasmic organelles, such as lysosomes surrounded by the nuclear envelope, were sometimes observed within one nucleus (*Figure 7C*). Crystalline inclusions were observed as lattice structures (*Figure 7D*). Some subcellular organelles, such as mitochondria, were frequently observed within the nucleus of *Dst-b^E2610Ter/E2610Ter* cardiomyocytes as a result of nuclear invaginations (*Figure 7E and F*) because they were surrounded by the nuclear membrane. Moreover, vacuoles that seemed to contain liquid were observed in the nucleus. In some cases, endoplasmic reticulum and ribosomes were observed at higher densities within the nucleus of *Dst-b^E2610Ter/E2610Ter* cardiomyocytes (*Figure 7G*), which is similar to the findings for the nucleoplasmic reticulum (*Bezin et al., 2008*; *Malhas et al., 2011*). Next, we investigated the molecular components of nuclear inclusions and found that the autophagy-associated protein p62 was deposited in *Dst-b^E2610Ter/E2610Ter* cardiomyocyte nuclei (*Figure 7H*). The percentage of cell nuclei harboring p62-positive structures was remarkably increased in *Dst-b^E2610Ter/E2610Ter* cardiomyocytes compared with that in WT cardiomyocytes (*Figure 7I*, WT mice: 0.55 ± 0.25% vs. *Dst-b^E2610Ter/E2610Ter* mice: 14.3% ± 1.9%; n = 3 WT mice: 871 total nuclei, n = 4 *Dst-b^E2610Ter/E2610Ter* mice: 860 total nuclei). We observed co-localization of p62 and ubiquitinated protein inside the nucleus

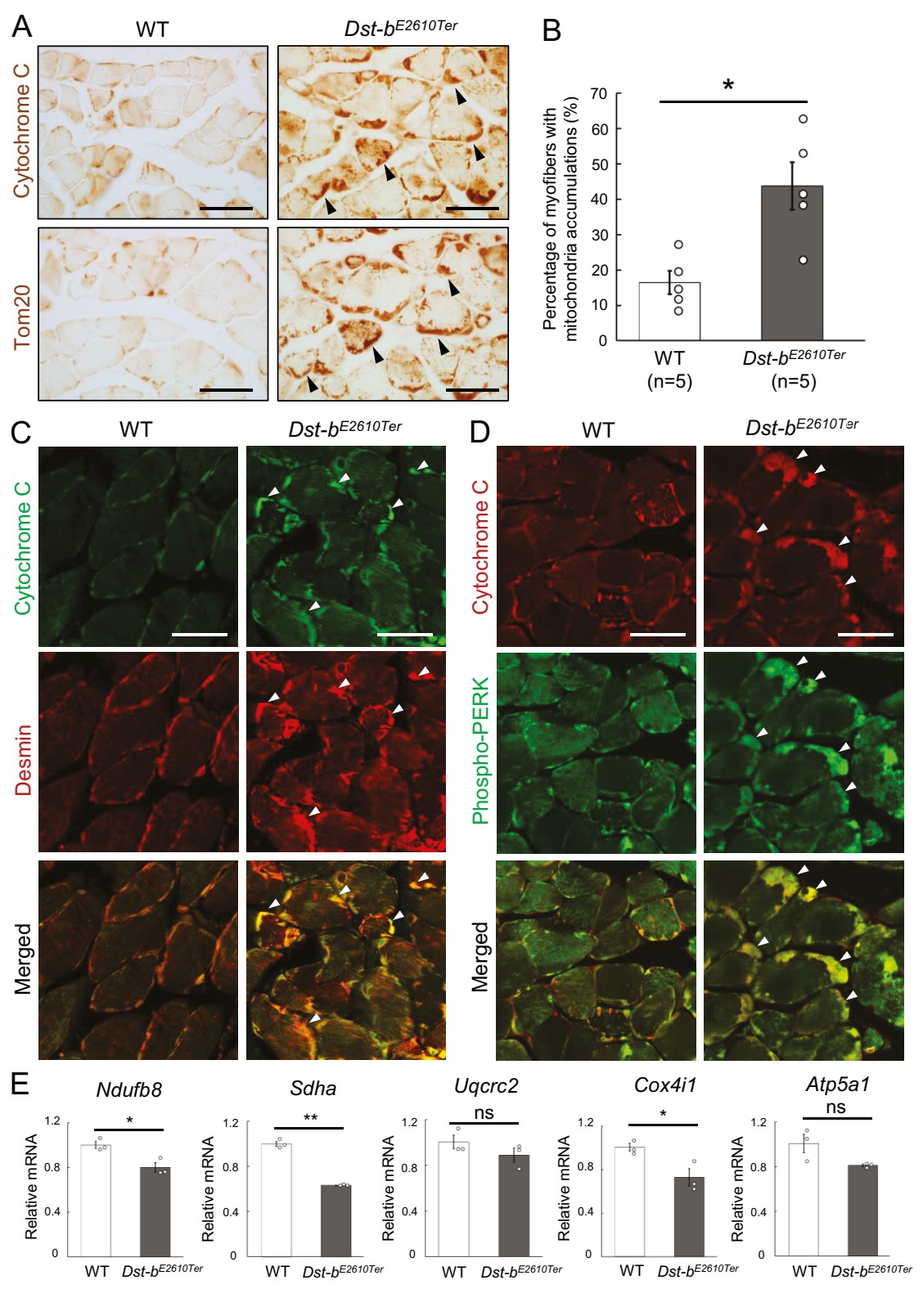

**Figure 5.** *Dst-b^E2610Ter* mutation leads mitochondrial alterations in skeletal muscle fibers. (**A**) Cytochrome C and Tom20 immunohistochemistry (IHC) on the serial cross sections of the soleus at 23 months of age. In WT soleus, mitochondria were mainly stained underneath the sarcolemma. *Dst-b^E2610Ter/E2610Ter* soleus showed accumulated mitochondria in subsarcolemmal regions (arrowheads). (**B**) Quantitative data of the percentage of myofibers with accumulated mitochondria in soleus of WT and *Dst-b^E2610Ter/E2610Ter* mice (n = 5 mice, each genotype, at 16–23 months old). * denotes statistically

*Figure 5 continued on next page*

Figure 5 continued

significant difference at p<0.05 (p=0.0109), using Student's *t*-test. (**C**) Double IHC of cytochrome C and desmin in cross sections of the soleus. Each protein was stained in subsarcolemmal spaces of WT soleus. In *Dst-b^(E2610Ter/E2610Ter)* soleus, cytochrome C-positive mitochondria were accumulated at the same positions with desmin aggregates (arrowheads). (**D**) Double IHC of cytochrome C and phospho-PERK. Phospho-PERK signals were strongly detected in subsarcolemmal regions in which cytochrome C-positive mitochondria accumulated (arrowheads). (**E**) qPCR analysis of genes responsible for oxidative phosphorylation in the soleus (n = 3 mice, each genotype, 21 months of age). * and ** denote statistically significant difference at p<0.05 (*Ndufb8*, p=0.0191; *Cox4i1*, p=0.0335) and p<0.01 (*Sdha*, p=0.0001), respectively, and ns means not statistically significant (*Uqcrc2*, p=0.2546; *Atp5a1*, p=0.0769), using Student's *t*-test. Data are presented as mean ± SE. Scale bars: (**A, C, D**) 50 μm.

using super-resolution microscopy (*Figure 7J* and *Video 1*). Small ubiquitin-related modifier (SUMO) proteins are major components of intranuclear inclusions in some neurological diseases, including neuronal intranuclear inclusion disease (NIID) (*Mori et al., 2012*; *Pountney et al., 2003*). SUMO proteins were also detected in the intranuclear structures of *Dst-b^(E2610Ter/E2610Ter)* cardiomyocytes using anti-SUMO-1 and anti-SUMO-2/3 antibodies (*Figure 7—figure supplement 1A and B*). Because desmin and αB-crystallin were absent from the intranuclear structures, protein aggregates in the nuclei were distinct from those in some molecular components of the cytoplasm (*Figure 7—figure supplement 1C and D*). The p62-positive structures inside nuclei were also devoid of LC3, which is an autophagy-associated molecule, whereas p62 and LC3 were co-localized in the cytoplasm (*Figure 7—figure supplement 1E*). Lamin A/C IHC also confirmed that the p62- and ubiquitin-positive structures were surrounded by or adjacent to lamin A/C-positive lamina (*Figure 7—figure supplement 1F and G*). Desmin IHC also confirmed cytoplasmic desmin in the space surrounded by highly invaginated nuclear membrane (*Figure 7—figure supplement 1H*). These results revealed that *Dst-b* gene mutations lead to dysmorphic nuclei and formation of nuclear inclusions containing p62, ubiquitinated proteins, and SUMO proteins in the *Dst-b^(E2610Ter/E2610Ter)* myocardium.

## Next-generation sequencing analysis of heart tissues from *Dst-b^(E2610Ter/E2610Ter)* mice

To determine the molecular etiology of pathophysiological features of *Dst-b^(E2610Ter/E2610Ter)* mouse hearts, we performed RNA-seq analysis of transcripts obtained from the ventricular myocardium of 14–19-month-old *Dst-b^(E2610Ter/E2610Ter)* mice and age-matched WT mice. Principal component analysis (PCA) and hierarchical clustering of RNA-seq data showed that transcriptomic characteristics between WT and *Dst-b^(E2610Ter/E2610Ter)* mouse hearts were different (*Figure 8—figure supplement 1A and B*). RNA-seq identified 728 differentially expressed genes with a false discovery rate (FDR) < 0.1 and p<0.001 (*Figure 8A*), including 481 upregulated genes and 247 downregulated genes in *Dst-b^(E2610Ter/E2610Ter)* mouse hearts compared with those from WT mice. A volcano plot demonstrated that upregulated genes were associated with pro-fibrotic pathways (e.g., *Ctgf*, *Tgfb2*, *Crlf1*), cytoskeletal regulation (e.g., *Acta2*, *Cenpf*, *Ankrd1*, *Mybpc2*, *Myom2*, *Nefm*), and metabolic control (*Ucp2*, *Nmrk2*, *Tbx15*) and that downregulated genes were associated with transmembrane ion transport (e.g., *Scn4b*, *Cacng6*, *Ano5*, *Tmem150c*, *Cacna1s*, *Scn4a*). The reliability of RNA-seq data was validated by using qPCR analysis (*Figure 8—figure supplement 1C*). The differentially expressed genes included several causative genes of congenital heart defects, hypertrophic cardiomyopathy, congenital long-QT syndrome, muscular dystrophy, and dilated cardiomyopathy (e.g., *Acta2*, *Ankrd1*, *Scn4b*, *Ano5*, *Rpl3l*, *Cenpf*) (*Figure 8B*). Gene Ontology (GO) analysis confirmed the following terms specific to the skeletal muscle and heart: biological process (e.g., upregulated: muscle contraction, heart development, skeletal muscle tissue development, sarcomere organization, and regulation of heart rate; downregulated: muscle organ development) and cellular components (e.g., upregulated: Z-disc, sarcolemma, myofibril, neuromuscular junctions, sarcoplasmic reticulum) (*Figure 8C and D*, *Supplementary file 1A–E*). Kyoto Encyclopedia of Genes and Genomics (KEGG) pathway analysis also indicated abnormal muscle pathways (e.g., upregulated: dilated cardiomyopathy, hypertrophic cardiomyopathy; downregulated: cardiac muscle contraction) (*Figure 8E*, *Supplementary file 1C and F*). These data provide a comprehensive landscape for cardiomyopathy in *Dst-b^(E2610Ter/E2610Ter)* mice.

## Mosaic analysis using conditional *Dst^Gt* mice

Because truncated Dst-b proteins expressed from the *Dst-b^(E2610Ter)* allele harbor an actin-binding domain (ABD) and a plakin domain, it is possible that they have a gain-of-function effect on the

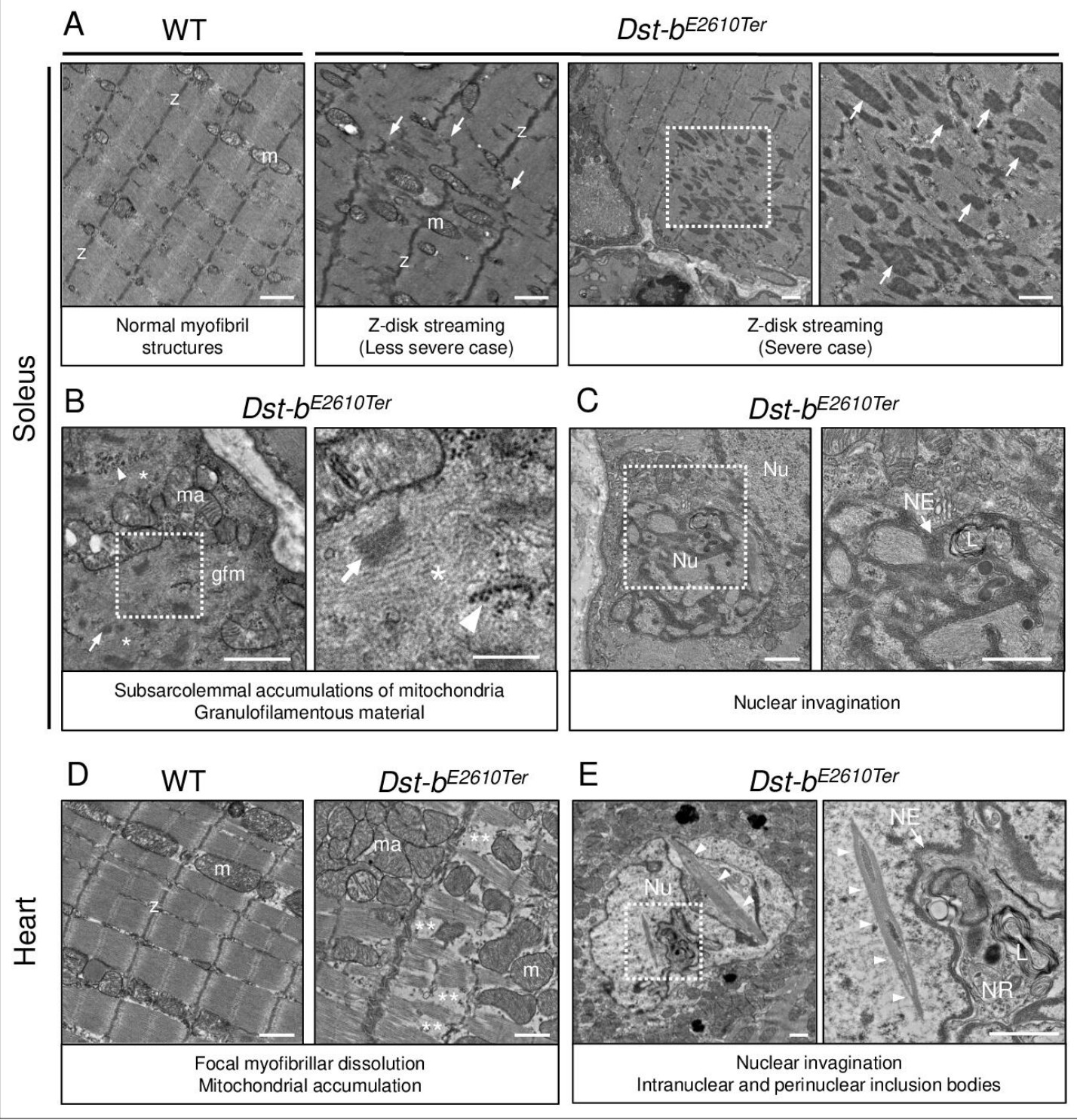

**Figure 6.** Transmission electron microscopy (TEM) analyses on the soleus and heart of *Dst-b^E2610Ter^* mice. (**A–C**) TEM images on longitudinal soleus ultrathin sections of WT and *Dst-b^E2610Ter/E2610Ter^* mice at 22 months of age (n = 2 mice, each genotype). (**A**) Normal myofibril structure with well-aligned Z-disks (z) and mitochondria (m) in WT soleus. In *Dst-b^E2610Ter/E2610Ter^* soleus, Z-disk streaming (arrows) was observed as displacements of Z-disks (less severe case) and focal myofibrillar dissolution (severe case). Dotted box in *Dst-b^E2610Ter/E2610Ter^* soleus indicates the severely disrupted myofibrillar structures associated with abnormally thickened Z-disks (arrows) shown as the inset. (**B**) Subsarcolemmal accumulations of mitochondria and granulofilamentous material (gfm) in *Dst-b^E2610Ter/E2610Ter^* soleus. Gfm consists of glycogen granule deposits (arrowheads) and filamentous materials with electron-dense Z-disk streaming (arrows) and filamentous pale area (asterisk). Accumulated mitochondria (ma) are adjacent to the gfm. Dotted box shows the enlarged structure of gfm. (**C**) Nuclear invaginations in *Dst-b^E2610Ter/E2610Ter^* soleus. Dotted box in the cell nucleus (Nu) indicates the deeply invaginated nuclear

*Figure 6 continued on next page*

*Figure 6 continued*

envelope (NE, arrow) involving lysosomes (L) shown as the inset. (**D, E**) TEM images on heart ultrathin sections (n = 2 mice, each genotype). (**D**) Normal myofibril structure with well-aligned Z-disks (z) and mitochondria (m) in WT cardiomyocytes (left panel). In *Dst-b*$^{E2610Ter/E2610Ter}$ cardiomyocytes, focal myofibrillar dissolution (double asterisks) was observed and accumulated mitochondria (ma) were evident there (right panel). (**E**) Abnormal shape of nucleus (Nu) and intranuclear inclusions were frequently observed in *Dst-b*$^{E2610Ter/E2610Ter}$ cardiomyocytes. Dotted box indicates that the NE (arrow) deeply invaginated and involved cytoplasmic components such as lysosomes (L) to form nucleoplasmic reticulum (NR) shown as inset. Dysmorphic nucleus harbored the crystalline inclusions in intranuclear and perinuclear regions (arrowheads). Scale bars: (**A-E**) 1µm; (**B** (right magnified images)) 400 nm.

pathogenesis of *Dst-b*$^{E2610Ter/E2610Ter}$ mice. Therefore, we analyzed conditional *Dst*$^{Gt}$ mice (*Figure 9A*), in which both Dst-a and Dst-b are trapped within the N-terminal ABD and lose their function in a genetically mosaic manner. We crossed female *Dst*$^{Gt-inv/Gt-inv}$ mice with male *β-actin* (*Actb*)-iCre; *Dst*$^{Gt-DO/wt}$ mice to generate *Actb-iCre; Dst*$^{Gt-DO/Gt-inv}$ mice (mosaic mice, *Figure 9B*). More than half of the mosaic mice survived over several months and showed mild *dt* phenotypes, such as smaller body size and impairment of motor coordination (*Figure 9C*). Western blotting analysis indicated that the Dst-b protein was almost absent in cardiac extract from surviving mosaic mice; however, the Dst-a protein was detected at variable levels in brain extract (*Figure 9D*). Cardiac fibrosis was observed in hearts of 10-month-old mosaic mice (*Figure 9E*). In addition, increases in *Nppa* mRNA expression and p62 deposition were evident in the left ventricular myocardium of mosaic mice compared with control mice (*Figure 9F*). These data suggest that myopathy in *Dst-b*$^{E2610Ter/E2610Ter}$ mice is caused by a loss-of-function mutation of *Dst-b*.

### *DST* mutant alleles with nonsense mutations in *DST-b*-specific exons

Next, we performed in silico screening for *DST* mutant alleles with nonsense mutations within *DST-b*-specific exons in a database containing normal human genomes. In a search of the dbSNP database, which contains information on single-nucleotide variations in healthy humans, we identified 58 different types of nonsense mutations (94 alleles total) in all five *DST-b*-specific exons (*Table 1*, *Figure 10A*). The alleles with nonsense mutations were found in different populations (European, Asian, African, and American, *Figure 10B*). In Japan, four alleles with nonsense mutations were found among 16,760 alleles in the ToMMo 8.3KJPN database (*Tadaka et al., 2021*), suggesting that homozygous or compound heterozygous mutants may exist in one per approximately 16 million people. We further investigated *DST-b* mutations in Japanese patients diagnosed with myopathy (*Nishikawa et al., 2017*). Although nonsense mutations were not identified in those patients, we identified two patients with myopathy who harbored compound heterozygous *DST* variants (*Supplementary file 1G*). All variants were predicted to be variants of uncertain significance (VUS); therefore, the involvement of these variants in myopathic manifestations needs to be carefully interpreted. Taken together, these data suggest that unidentified familial myopathies and/or cardiomyopathies caused by *DST-b* mutant alleles exist in all populations worldwide.

## Discussion

In this study, we established the isoform-specific *Dst-b* mutant mouse as a novel animal model for late-onset protein aggregate myopathy. Myopathic alterations in *Dst-b* mutants were similar to those observed for MFMs. We also found nuclear inclusion as a unique pathological hallmark of *Dst-b* mutant cardiomyocytes, which provided molecular insight into pathophysiological mechanisms of cardiomyopathy. Because the *Dst-b*-specific nonsense mutation resulted in myopathy without sensory neuropathy, and mutant *DST* alleles with nonsense mutations in *DST-b*-specific exons exist, our data suggest that unidentified human myopathy may be caused by these *DST-b* mutant alleles. Our data also indicated that the sensory neurodegeneration, movement disorders, and lethality in *dt* mice and patients with HSAN-VI, which are caused by mutations of both the *Dst-a* (*DST-a*) and *Dst-b* (*DST-b*) isoforms, are actually caused by deficiency of the *Dst-a* (*DST-a*) isoform.

### Isoform-specific *Dst-b* mutants result in late-onset protein aggregate myopathy

Isoform-specific *Dst-b* mutant mice exhibited late-onset protein aggregate myopathy with cardiomyopathy, which could be termed 'dystoninopathy.' Protein aggregates observed in the striated

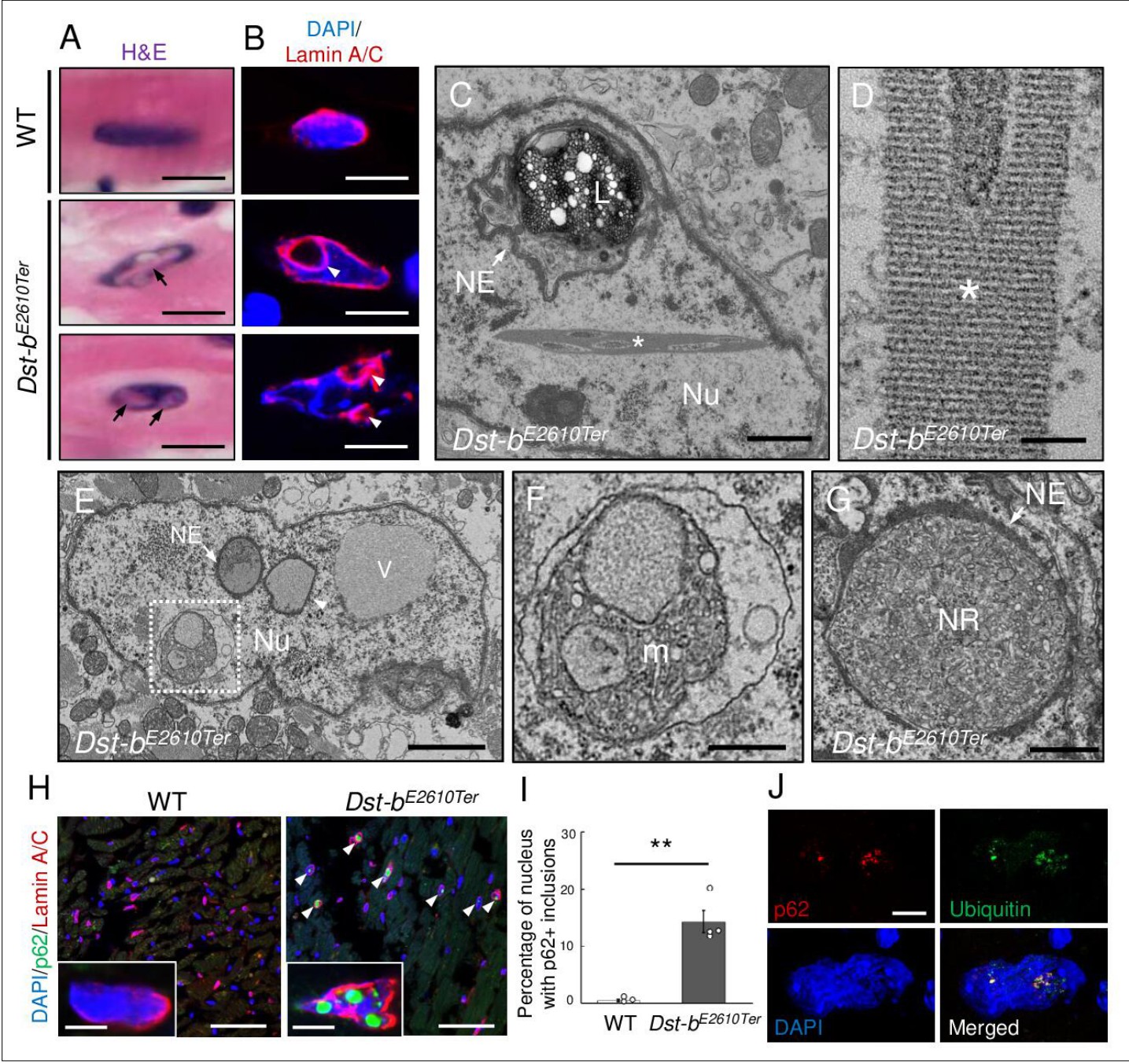

**Figure 7.** Defects in nuclear structure and intranuclear inclusions in cardiomyocytes of *Dst-b^E2610Ter* mice. (**A**) H&E staining of cardiomyocyte nuclei. Eosinophilic structures (arrows) were observed inside the nucleus of *Dst-b^E2610Ter/E2610Ter* cardiomyocytes. (**B**) Nuclear lamina was immunolabeled with anti-lamin-A/C antibody. In dysmorphic nuclei of *Dst-b^E2610Ter/E2610Ter* cardiomyocytes, invaginated nuclear lamina did not contain DAPI signals (arrowheads). (**C–G**) Transmission electron microscope (TEM) images of cardiomyocyte nuclei of *Dst-b^E2610Ter/E2610Ter* mice (n = 2 mice, each genotype). (**C**) Crystalline inclusions (asterisk) and cytoplasmic organelles such as lysosomes (L) surrounded by the nuclear envelope (NE, arrow) were observed inside nucleus (Nu). (**D**) Crystalline inclusions are lattice structure. (**E**) The NE surrounded various components such as organelles (arrow) or presumably liquid (arrowheads). There was a vacuole (V) seemed to contain liquid. (**F**) Dotted box (**E**) indicates mitochondria (m) surrounded by a membrane. (**G**) An organelle surrounded by the NE (arrow) contained endoplasmic reticulum and ribosomes. Such structure was discriminated as nucleoplasmic reticulum (NR). (**H**) Immunofluorescent images of heart sections labeled with anti-p62 and anti-lamin A/C antibodies. p62 was deposited inside nuclei surrounded by lamin A/C-positive lamina in *Dst-b^E2610Ter/E2610Ter* cardiomyocytes (arrowheads). (**I**) Quantitative data representing percentages of cardiomyocyte nuclei harboring p62-positive inclusions (n = 3 WT mice; n = 4 *Dst-b^E2610Ter/E2610Ter* mice). ** denotes statistically significant difference at p<0.01 (p=0.0062), using Student's *t*-test. Data are presented as mean ± SE. (**J**) Super-resolution microscopy images showed co-localization of p62 and ubiquitin in the nucleus

*Figure 7 continued on next page*

*Figure 7 continued*

of *Dst-b*$^{E2610Ter/E2610Ter}$ cardiomyocytes. Immunohistochemistry (IHC) analysis of molecular features of nuclear inclusions are shown in *Figure 7—figure supplement 1*. Scale bars: (**A, B, H** (insets), **J**) 5µm; (**C, G**) 1µm; (**D**) 100nm; (**E**) 2µm; (**F**) 600 nm; (**H**) 50µm.

The online version of this article includes the following figure supplement(s) for figure 7:

**Figure supplement 1.** Histological features of intranuclear structures in *Dst-b*$^{E2610Ter/E2610Ter}$ cardiomyocytes.

muscles of *Dst-b* mutant mice were composed of desmin, αB-crystallin, plectin, and truncated Dst-b. MFMs are major groups of protein aggregate myopathies and are caused by mutations of genes encoding Z-disk-associated proteins (e.g., *DES, CRYAB, MYOT, ZASP, FLNC, BAG3, PLEC*) (*Keduka et al., 2012*; *Clemen et al., 2013*; *Batonnet-Pichon et al., 2017*). It is plausible that Dst proteins are also aggregated in muscles affected by other MFM types. Because Dst-b encodes cytoskeletal linker proteins that are localized in the Z-disks, it is possible that mutations in *Dst-b* lead to Z-disk fragility, and repeated contraction may lead to myofibril disruption and induction of muscle regeneration. In line with this notion, we observed Z-disk streaming and CNFs, which reflect muscle degeneration and regeneration.

It is possible that protein quality control impairment through the unfolded protein response, autophagy, and the ubiquitin proteasome system is involved in protein aggregation in *Dst-b* mutant mouse muscles. Moreover, several genes responsible for the unfolded protein response, including heat shock proteins, were upregulated in *Dst-b* mutant mouse hearts. We also observed the accumulation of the autophagy-associated molecules p62 and ubiquitinated proteins in the muscles of *Dst-b* mutant mice. Co-aggregation of p62 and ubiquitinated proteins has been reported in autophagy-deficient neurons of *Atg7* conditional knockout mice (*Komatsu et al., 2007*). In the sensory neurons of *dt* mice, loss of *Dst* was reported to lead to disruption of the autophagic process (*Ferrier et al., 2015*; *Lynch-Godrei et al., 2021*). Autophagy was also reported to be involved in the pathogenesis of other types of MFMs; the *CryAB R120G* mutation (*CryAB*$^{R120G}$) increases autophagic activity, and genetic inactivation of autophagy aggravates heart failure in *CryAB*$^{R120G}$ mice (*Tannous et al., 2008*), while enhanced autophagy ameliorates protein aggregation in *CryAB*$^{R120G}$ mice (*Bhuiyan et al., 2013*). It would be interesting to investigate the involvement of autophagy and other protein degradation machinery in the process of protein aggregation myopathy in *Dst-b*$^{E2610Ter/E2610Ter}$ mutant mice.

Accumulated evidence strongly suggests that myofibrillar integrity is essential for the maintenance of mitochondrial function (*Vincent et al., 2016*). Mitochondrial dysfunctions have also been well recognized in some animal models of MFMs, such as desmin (*Des*), αB-crystallin (*Cryab*), and plectin (*Plec*) mutants (*Winter et al., 2015*; *Winter et al., 2016*; *Diokmetzidou et al., 2016*; *Alam et al., 2020*). Abnormal accumulation of mitochondria in the subsarcolemmal space was observed in *Dst-b*$^{E2610Ter/E2610Ter}$ mutant muscles. Mitochondrial dysfunction in *Dst-b*$^{E2610Ter/E2610Ter}$ mutant mouse cardiomyocytes was also suggested by RNA-seq data, including altered expression of genes responsible for oxidative phosphorylation and other metabolic processes. The altered heart functions characterized by long QT intervals and PVCs in *Dst-b* mutant mice may be explained by mitochondrial dysfunction, as well as altered expression of genes involved in transmembrane ion transport and muscle contraction.

## Nuclear abnormalities in *Dst-b* mutation-induced myopathy and cardiomyopathy

Deep invaginations of the nuclear membrane were also a characteristic feature of *Dst-b* mutant muscle fibers. The linker of nucleoskeleton and cytoskeleton (LINC) complex is a protein complex that spans the inner and outer nuclear membrane to bridge the cytoskeleton in the cytoplasm to the nuclear lamina underlying the inner nuclear membrane (*Stroud et al., 2014*). Mutations in genes encoding LINC complex proteins are

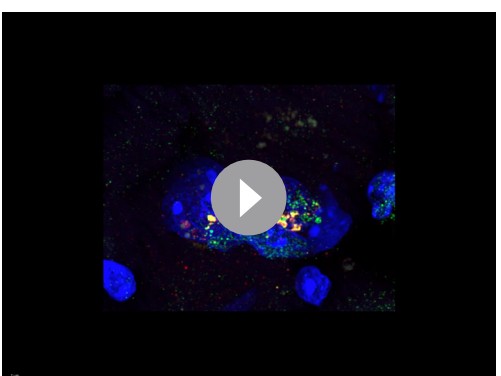

**Video 1.** Super-resolution microscopy image showing co-localization of p62 and ubiquitin in the nucleus of *Dst-b* mutant cardiomyocytes.

https://elifesciences.org/articles/78419/figures#video1

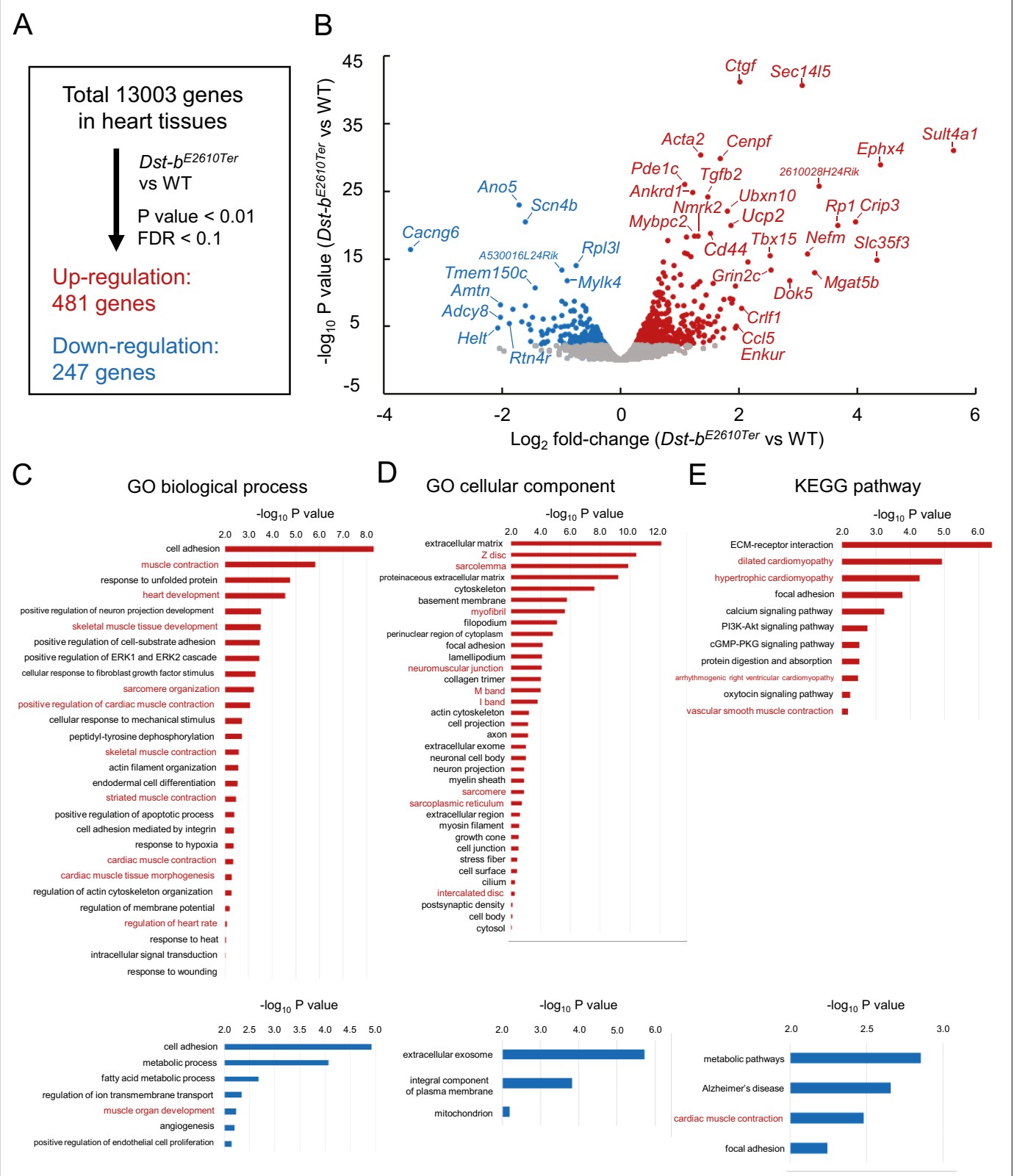

**Figure 8.** RNA-seq-based transcriptome of hearts from *Dst-b^E2610Ter* mice. RNA-seq analysis was performed in ventricular myocardium tissues from 14- to 19-month-old *Dst-b^E2610Ter/E2610Ter* mice and WT mice (n = 3, each genotype). Principal component analysis (PCA) and hierarchical clustering of RNA-seq data are shown in *Figure 8—figure supplement 1*. (**A**) Among the 13,003 genes expressed in the heart, 481 genes were upregulated and 247 genes were downregulated. Thresholds were set at p<0.01 and false discovery rate (FDR) < 0.1, respectively. (**B**) Volcano plot shows differentially expressed

*Figure 8 continued on next page*

*Figure 8 continued*

genes. Red and blue dots represent genes upregulated and downregulated, respectively. Dots of highly changed genes were labeled with gene symbols. Changes of gene expressions were validated by qPCR in *Figure 8—figure supplement 1*. Bar graphs show Gene Ontology (GO) biological process (**C**), GO cellular component (**D**), and Kyoto Encyclopedia of Genes and Genomics (KEGG) pathway (**E**) enriched in genes that are upregulated and downregulated in the heart of *Dst-b*$^{E2610Ter/E2610Ter}$ mice. Red and blue bars represent genes upregulated and downregulated, respectively. Items in red words are specific to skeletal muscle and heart. List of genes resulting from GO analysis and KEGG pathway analysis is shown in *Supplementary file 1A–F*. RNA-seq data can be accessed from the Gene Expression Omnibus under accession # GSE184101.

The online version of this article includes the following figure supplement(s) for figure 8:

**Figure supplement 1.** Transcriptomes of hearts from WT and *Dst-b*$^{E2610Ter}$ mice.

known to cause skeletal myopathy and cardiomyopathy, called Emery–Dreifuss muscular dystrophy (*Heller et al., 2020*), and defects in nuclear structure are often observed in mice with mutations of LINC and LINC-associated molecules such as lamin A (*Lmna*), nesprin, ezrin, and *Des* (*Nikolova et al., 2004*; *Banerjee et al., 2014*; *Wada et al., 2019*; *Heffler et al., 2020*). Because Dst can associate with nesprin-3α via the ABD (*Young and Kothary, 2008*), and we demonstrated that Dst-b can localize around nuclei of muscle fibers, Dst-b may be involved in the molecular bridge between the LINC complex and cytoskeleton and maintain the shape of nuclei.

Furthermore, we found that nuclear inclusion was a unique hallmark of cardiomyopathy in *Dst-b* mutant mice. The nuclear inclusions were different from the cytoplasmic protein aggregates in autophagy-deficient mice because nuclear inclusions lacked LC3 protein. Intranuclear inclusions are also observed in NIID, which is a neurological disease caused by GGC repeat expansion of the *NOTCH2NLC* gene (*Sone et al., 2019*), which has also been reported to be associated with oculo-pharyngodistal myopathy with intranuclear inclusions (*Ogasawara et al., 2020*). The intranuclear inclusions of NIID predominantly include p62, ubiquitinated proteins, and SUMOylated proteins (*Mori et al., 2012*; *Pountney et al., 2003*). In our observations, SUMOylated proteins were also present in the nuclear inclusions of *Dst-b* mutants. Thus, nuclear inclusions of *Dst-b* mutant cardio-myocytes have common features with those of NIID in terms of molecular components (*Sone et al., 2011*). It would be interesting to further investigate the molecular components and pathogenesis of nuclear inclusions in the cardiomyopathy of *Dst-b* mutant mice and compare them with those found in NIID.

## Role of the Dst-b isoform in *DST*-related diseases

A wide variety of genetic diseases have been reported to be caused by *DST* mutations. Loss-of-function mutations in both *DST-a* and *DST-b* result in HSAN-VI, and loss-of-function mutations of *DST-e* result in the skin blistering disease epidermolysis bullosa simplex (*Groves et al., 2010*; *Edvardson et al., 2012*). Recently, *DST* was reported as a candidate gene of pulmonary atresia, a rare congenital heart defect (*Shi et al., 2020*). According to this study, it is possible that unidentified hereditary myopathies and heart disease caused by human *DST-b*-specific mutations exist. Because the causative gene is unidentified in almost half of MFM cases, it would be worthwhile to perform candidate gene screening on *DST-b*-specific exons in patients with MFMs.

Recently, HSAN-VI has been proposed to be a spectrum disease because of its varying disease severity and the diverse deficiency of DST isoforms (*Lynch-Godrei and Kothary, 2020*). DST-a2 seems to be a crucial isoform for HSAN-VI pathogenesis because *DST-a2*-specific mutations result in adult-onset HSAN-VI (*Manganelli et al., 2017*; *Fortugno et al., 2019*), and neuronal expression of *Dst-a2* partially rescues the *dt* phenotype (*Ferrier et al., 2014*). We suggest that *Dst-a1* and *Dst-a2* play redundant roles because *Dst*$^{Gt}$ mice, in which both *Dst-a1* and *Dst-a2* are trapped, show a severe phenotype (*Horie et al., 2014*). Among three *Dst-b* isoforms, remarkable reduction of *Dst-b1*, but not *Dst-b2* or *Dst-b3*, was observed in *Dst-b*$^{E2610Ter/E2610Ter}$ mice, suggesting that reduced *Dst-b1* expression is involved in myopathic phenotypes in *Dst-b*$^{E2610Ter/E2610Ter}$ mice. It is possible that DST isoforms 1 and 2 may be important to different extents in neural and muscular tissues. Regarding this point of view, it would be interesting to compare muscle phenotypes between patients harboring *DST-a1*, *-a2*, *-b1*, and *-b2* isoform mutations and patients with *DST-a2* and *-b2* isoform-specific mutations (*Manganelli et al., 2017*; *Fortugno et al., 2019*; *Motley et al., 2020*).

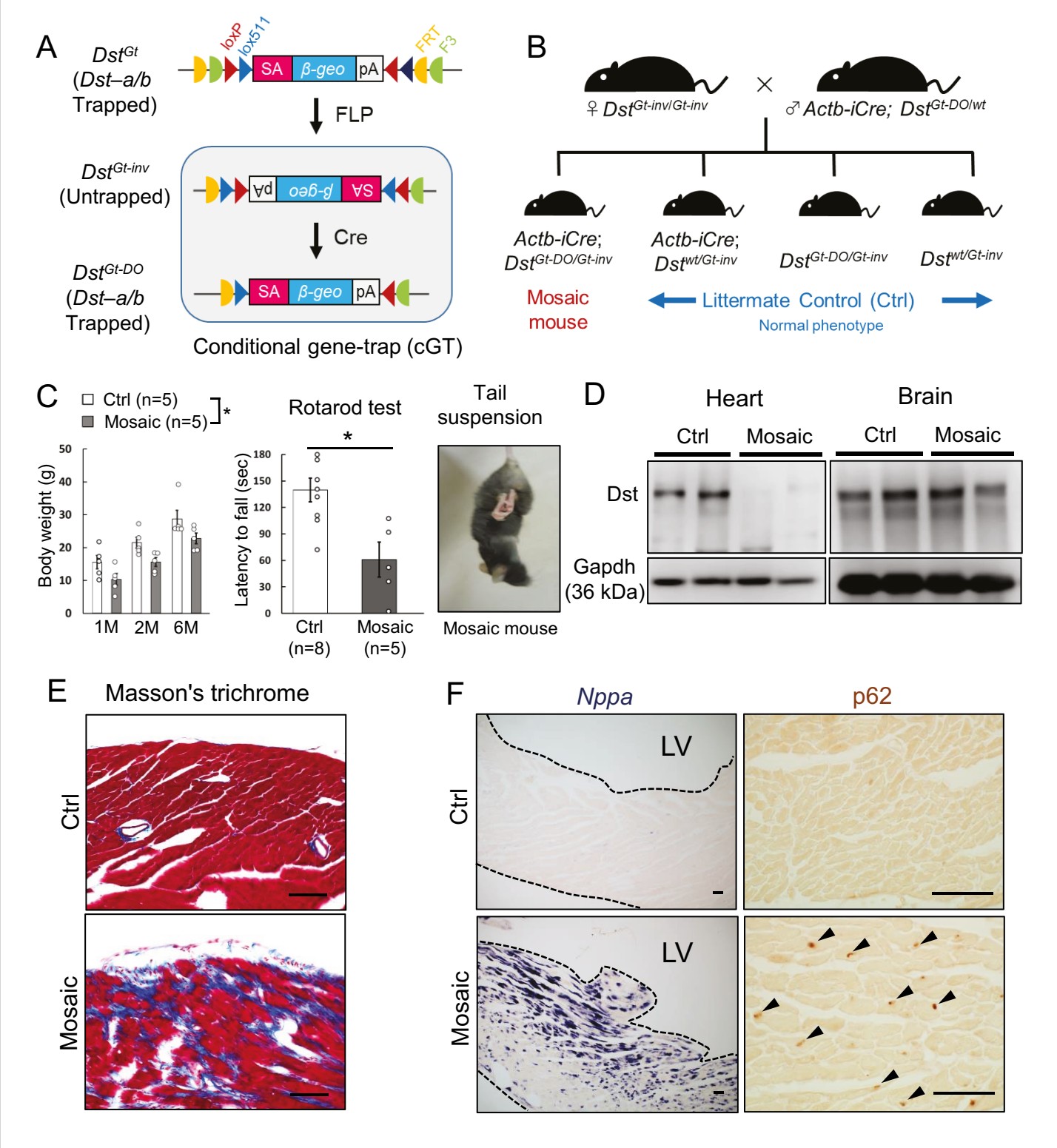

**Figure 9.** Cardiomyopathy in conditional *Dst* conditional gene trap (cGT) mice. (**A**) Scheme of mosaic analysis by cGT of *Dst-a/b*. The gene trap cassette contains splice acceptor (SA) sequence, the reporter gene βgeo, and poly-A (pA) termination signal. The gene trap cassette is flanked by pairs of inversely oriented target sites of FLP recombinase (Frt and F3: half circles) and Cre recombinase (loxP and lox5171: triangles). FLP- and Cre-mediated recombination induce irreversible inversion from mutant *Dst^Gt* allele to untrapped *Dst^Gt-inv* allele and *Dst^Gt-inv* allele to mutant *Dst^Gt-DO* allele, respectively. (**B**) For generation of *Actb-iCre; Dst^Gt-DO/Gt-inv* (mosaic) mice, female *Dst^Gt-inv/Gt-inv* mice were mated with male *Actb-iCre; Dst^Gt-DO/WT* mice. (**C**) Body weight of

*Figure 9 continued on next page*

*Figure 9 continued*

male mosaic mice reduced than Ctrl mice in several months of age (n = 5 mice, each genotype; two-way ANOVA; genotype effect: p=0.0493; age effect: p=0.0000; genotype × age interaction: p=0.9126). Impairment of motor coordination in mosaic mice is shown by rotarod test (n = 8 Ctrl mice, n = 5 mosaic mice). * denotes statistically significant difference at p<0.05 (p=0.0110), using Student's *t*-test. Mosaic mice displayed hindlimb clasping and twist movements during tail suspension. (**D**) Representative data of Western blot analysis showed a deletion of Dst band in heart lysates from mosaic mice, while residual Dst bands were detected in those brain lysates (n = 2 mice, each genotype, 5–7 months of age). (**E**) Masson's trichrome staining showed extensive fibrosis in heart sections of mosaic mice at 10 months of age. (**F**) *Nppa* mRNA and p62-positive depositions (arrowheads) were evident in the left ventricular myocardium of mosaic mice than Ctrl mice. Data are presented as mean ± SE. Scale bars: (**E, F**) 50 µm.

## Roles of plakin family proteins in muscles and other tissues

Members of the plakin protein family other than Dst, such as plectin and desmoplakin, are known to be expressed and play crucial roles in maintaining muscle integrity (*Leung et al., 2002*; *Boyer et al., 2010b*; *Horie et al., 2017*). Plectin is a Dst-associated protein and one of the most investigated members of the plakin protein family (*Castañón et al., 2013*). There are four alternative splicing isoforms of plectin in striated muscle fibers, which are localized in differential subcellular compartments: plectin 1 in nuclei, plectin 1b in mitochondria, plectin 1d in Z-disks, and plectin 1f in the sarcolemma (*Mihailovska et al., 2014*; *Staszewska et al., 2015*; *Winter et al., 2015*). *Plec* deficiency is lethal in mice at the neonatal stage, and these mice exhibit skin blistering and muscle abnormalities (*Andrä et al., 1997*). Conditional deletion of *Plec* in muscle fibers leads to progressive pathological alterations, such as aggregation of desmin and chaperon protein, subsarcolemmal accumulation of mitochondria, and an abnormal nuclear shape (*Konieczny et al., 2008*; *Staszewska et al., 2015*; *Winter et al., 2014*; *Winter et al., 2015*), some of which are also observed in the muscles of aged *Dst-b* mutant mice.

Dst and plectin also have common features in other cell types. In keratinocytes, plectin and Dst-e localize to the inner plaque of hemidesmosomes and anchor keratin intermediate fibers to hemidesmosomes (*Künzli et al., 2016*). Conditional deletion of *Plec* from epidermal cells causes epidermal barrier defects and skin blistering (*Ackerl et al., 2007*), both of which is more severe than that observed in *dt* mice carrying *Dst-e* mutations (*Guo et al., 1995*; *Yoshioka et al., 2020*). Furthermore, we recently demonstrated that conditional deletion of *Dst* from Schwann cells in the peripheral nervous system leads to disorganization of the myelin sheath (*Horie et al., 2020*), similar to that observed for *Plec*-deficient Schwann cells (*Walko et al., 2013*). These studies suggest that Dst and plectin, which are plakin family proteins, have overlapping roles in different cell types.

## Materials and methods

### Animals

*Dst-b*^E2610Ter^ mice, *Dst*^Gt(E182H05)^ mice (MGI number: 3917429; *Horie et al., 2014*), and *Actb-iCre* mice (*Zhou et al., 2018*) were used in this study. *Dst*^Gt(E182H05)^ allele was abbreviated as *Dst*^Gt^. Homozygous *Dst-b*^E2610Ter^ and *Dst*^Gt^ mice were obtained by heterozygous mating. For mosaic analysis, female *Dst*^Gt-inv/Gt-inv^ mice were crossed with male *Actb-iCre;Dst*^Gt-DO/wt^ mice. *Dst-b*^E2610Ter^ mutant line was C57BL/6J. The *Dst*^Gt^ mutant line was backcrossed to C57BL/6NCrj at least 10 generations. Mice were maintained in groups at 23°C ± 3°C, 50% ± 10% humidity, 12 hr light/dark cycles, and food/water availability ad libitum. Both male and female mice were analyzed in this study. Genotyping PCR for the *Dst*^Gt^ allele was performed as previous described (*Horie et al., 2014*). For genotyping PCR of the *Dst-b*^E2610Ter^ allele, the following primer set was used to amplify 377 bp fragments: *Dst-b* forward: 5′-TGA GCG ATG GTA GCG ACT TG-3′ and *Dst-b* reverse: 5′-GCG ACA CAC CTT TAG TTG CC-3′. PCR was performed using Quick Taq HS DyeMix (Toyobo, Osaka, Japan) and a PCR Thermal Cycler Dice (TP650; Takara Bio Inc, Shiga, Japan), with the following cycling conditions: 94°C for 2 min, followed by 32 cycles of 94°C for 20 s, 61°C for 30 s, and 72°C for 30 s. PCR products were cut with *Xho*I, which produced two fragments (220 bp and 157 bp) for the mutant *Dst-b*^E2610Ter^ allele. For genotyping PCR for *Actb-iCre* knockin allele, the following primer set was used to amplify 540 bp fragments: *iCre-F*: CTC AAC ATG CTG CAC AGG AGA T-3′ and *iCre-R*: 5′-ACC ATA GAT CAG GCG GTG GGT-3′.

**Table 1.** List of identified nonsense mutations in *DST-b*.

| No. | SNP_ID | *DST-b*-specific exon (Ex40-Ex44) | Base substitution (NM_001374736.1) | Amino acid substitution (NP_001361651.1) | Database | Global frequency | Specific population frequency |
|---|---|---|---|---|---|---|---|
| 1 | 780727375 | Ex40 | c.5513_5514insTTAGA, | p.Ser1839_Ly1a840insTer | ExAC | 1/120712 | European: 1/73330 |
| 2 | 775037762 | Ex40 | c.5581C>T | p.Gln1861Ter | ExAC | 1/120756 | European: 1/73346 |
| 3 | 267601090 | Ex40 | c.6106C>T | p.Gln2036Ter | None | None | None |
| 4 | 763489373 | Ex40 | c.6199C>T | p.Arg2067Ter | GnomAD_exome GnomAD ExAC ALFA | 5/248332 1/140170 1/120410 0/10680 | Asian: 3/48548; American: 2/34434, European: 1/75902 American: 1/11450 None |
| 5 | 980428529 | Ex40 | c.6413T>A | p.Leu2138Ter | TOPMED ALFA | 1/264690 1/35428 | None European: 1/26584 |
| 6 | 2098511415 | Ex40 | c.6846T>A | p.Cy1b282Ter | ALFA | 2/21326 | European: 2/16854 |
| 7 | 1416256967 | Ex40 | c.6907G>T | p.Glu2303Ter | TOPMED ALFA | 1/264690 0/14050 | None None |
| 8 | 1563150977 | Ex40 | c.6952C>T | p.Gln2318Ter | GnomAD_exome | 1/248356 | European: 1/133910 |
| 9 | 2098509730 | Ex40 | c.7045C>T | p.Gln2349Ter | GnomAD ALFA | 1/140156 0/10680 | African: 1/42028 None |
| 10 | 536128073 | Ex40 | c.7120C>T | p.Arg2374Ter | TOPMED GnomAD_exome GnomAD ALFA KOREAN GoNL | 1/264690 3/222812 1/139940 3/32028 1/2922 1/998 | None European: 3/118158 African: 1/41914 European: 1/23832; Other: 2/4554 KOREAN: 1/2922 None |
| 11 | 747917821 | Ex40 | c.7171C>T | p.Gln2391Ter | GnomAD ExAC | 1/140100 2/83458 | African: 1/41990 Asian: 1/19128; African: 1/7454 |
| 12 | 1261702898 | Ex40 | c.7214C>G, c.7214C>A, | p.Ser2405Ter | TOPMED GnomAD_exome ALFA | 1/264690 (C>A) 1/245012 (C>G) 0/10680 (C>A) | None European: 1/131624 None |
| 13 | 757004287 | Ex40 | c.7316T>G | p.Leu2439Ter | GnomAD_exome ExAC | 1/248436 1/120426 | Asian: 1/48550 Asian: 1/25104 |
| 14 | 559852499 | Ex40 | c.7459C>T | p.Gln2487Ter | GnomAD_exome ExAC 1000G | 1/247718 1/119556 1/5008 | Asian: 1/48546 Asian: 1/25084 South Asian: 1/978 |
| 15 | 1437052580 | Ex40 | c.7510C>T | p.Gln2504Ter | GnomAD_exome ALFA | 1/247692 1/8988 | Asian: 1/48526 Asian: 1/56 |
| 16 | 751368429 | Ex40 | c.7531_7534del | p.Leu2510_Asn2511insTer | GnomAD_exome ExAC | 1/247498 1/120056 | American: 1/34358 American: 1/11378 |
| 17 | 1563144635 | Ex40 | c.7447_7534del | p.Ile2482_Gly2483insTer | GnomAD_exome | 1/247498 | Asian: 1/48522 |
| 18 | 747767227 | Ex40 | c.7552C>T | p.Gln2518Ter | None | None | None |
| 19 | 1243608666 | Ex40 | c.7578C>G | p.Tyr2526Ter | GnomAD_exome | 1/247554 | European: 1/133118 |
| 20 | 1190095913 | Ex40 | c.7627C>T | p.Gln2543Ter | TOPMED GnomAD_exome ALFA | 1/264690 1/247988 0/10680 | None African: 1/15478 None |
| 21 | 756643045 | Ex40 | c.8014C>T | p.Gln2672Ter | TOPMED GnomAD_exome ExAC ALFA ALSPAC TWINSUK | 2/264690 2/248730 1/120578 0/14050 0/3854 1/3708 | None European: 2/134142 European: 1/73284 None None TWIN COHORT: 1/3708 |
| 22 | 1208663117 | Ex40 | c.8294G>A | p.Trp2765Ter | GnomAD_exome ALFA | 1/245286 1/21368 | European: 1/131776 European: 1/16886 |
| 23 | 2098496754 | Ex40 | c.8374G>T | p.Glu2792Ter | GnomAD ALFA | 1/140098 0/10680 | African: 1/42014 None |

*Table 1 continued on next page*

Table 1 continued

| No. | SNP_ID | DST-b-specific exon (Ex40-Ex44) | Base substitution (NM_001374736.1) | Amino acid substitution (NP_001361651.1) | Database | Global frequency | Specific population frequency |
|---|---|---|---|---|---|---|---|
| 24 | 1314301705 | Ex40 | c.8485C>T | p.Gln2829Ter, | TOPMED GnomAD ALFA | 1/264690 1/139898 0/14050 | None Ashkenazi Jewish: 1/3318 None |
| 25 | 1563133123 | Ex40 | c.8623del | p.Arg2874_ Val2875insTer | None | None | None |
| 26 | 2098493114 | Ex40 | c.8635C>T | p.Gln2879Ter | 8.3KJPN | 1/16760 | JAPANESE: 1/16760 |
| 27 | 1458968582 | Ex40 | c.8900C>G | p.Ser2967Ter | None | None | None |
| 28 | 1048157544 | Ex40 | c.9076C>T | p.Gln3026Ter | TOPMED ALFA | 1/264690 0/14050 | None None |
| 29 | 910403635 | Ex40 | c.9172G>T | p.Gly3058Ter | TOPMED ALFA | 2/264690 0/14050 | None None |
| 30 | 747173454 | Ex40 | c.9202G>T | p.Glu3068Ter | GnomAD_exome ExAC | 1/248348 1/120120 | European: 1/133958 European: 1/73002 |
| 31 | 751807675 | Ex40 | c.9227_9237del | p.Leu3075_ Leu3076insTer | ExAC | 1/119230 | European: 1/72390 |
| 32 | 749282620 | Ex40 | c.9439A>T | p.Ly1c147Ter | GnomAD ALFA | 1/140058 0/10680 | African: 1/42014 None |
| 33 | 1301999896 | Ex40 | c.9549C>G | p.Tyr3183Ter | TOPMED GnomAD_exome ALFA | 2/264690 1/247036 0/10680 | None European: 1/132884 None |
| 34 | 1411974489 | Ex40 | c.9580A>T | p.Ly1c194Ter | GnomAD_exome ALFA | 1/247422 1/8988 | Ashkenazi Jewish: 1/10018 European: 1/6062 |
| 35 | 2098482258 | Ex40 | c.9586del | p.Asp3195_ Val3196insTer | None | None | None |
| 36 | 972168431 | Ex40 | c.9818C>G, c.9818C>A | p.Ser3273Ter | TOPMED ALFA | 2/264690 (C>A) 0/14050 (C>A) | None None |
| 37 | 200867945 | Ex40 | c.9824T>A | p.Leu3275Ter | None | None | None |
| 38 | 1346974625 | Ex40 | c.10045C>T | p.Gln3349Ter | TOPMED GnomAD_exome ALFA | 1/264690 2/247418 1/33212 | None European: 2/133356 European: 1/24496 |
| 39 | 1229343851 | Ex40 | c.10114G>T | p.Glu3372Ter | None | None | None |
| 40 | 2098476192 | Ex40 | c.10166T>A | p.Leu3389Ter | TOPMED ALFA | 1/264690 0/10680 | None None |
| 41 | 2098474844 | Ex40 | c.10271C>A | p.Ser3424Ter, | TOPMED ALFA | 1/264690 0/10680 | None None |
| 42 | 2098473553 | Ex40 | c.10391del | p.Glu3463_ Leu3464insTer | TOPMED ALFA | 1/264690 0/10680 | None None |
| 43 | 1249289191 | Ex40 | c.10516G>T | p.Glu3506Ter | 8.3KJPN | 1/16760 | JAPANESE: 1/16760 |
| 44 | 1428617557 | Ex40 | c.10570G>T | p.Glu3524Ter | TOPMED GnomAD ALFA | 1/264690 1/139986 0/11862 | None European: 1/75810 None |
| 45 | 1586342297 | Ex40 | c.10633G>T | p.Glu3545Ter | KOREAN | 1/2922 | KOREAN: 1/2922 |
| 46 | 2098467261 | Ex41 | c.10807C>T | p.Gln3603Ter | GnomAD | 2/140068 | European: 2/75876 |
| 47 | 1586330106 | Ex41 | c.10815T>A | p.Cy1c605Ter | Korea1K | 1/1832 | KOREAN: 1/1832 |
| 48 | 2098463511 | Ex42 | c.10958_10959insTTA | p.Leu3653delinsPheTer | ALFA | 0/11862 | None |
| 49 | 2098462327 | Ex42 | c.11071C>T | p.Gln3691Ter | GnomAD ALFA | 1/139940 0/10680 | American: 1/13576 None |
| 50 | 1245541628 | Ex43 | c.11204C>G | p.Ser3735Ter | GnomAD ALFA | 1/139898 0/10680 | European: 1/75804 None |

Table 1 continued

| No. | SNP_ID | *DST-b*-specific exon (Ex40-Ex44) | Base substitution (NM_001374736.1) | Amino acid substitution (NP_001361651.1) | Database | Global frequency | Specific population frequency |
|---|---|---|---|---|---|---|---|
| 51 | 1363675987 | Ex43 | c.11210C>G | p.Ser3737Ter | TOPMED<br>GnomAD<br>ALFA | 1/264690<br>1/139886<br>0/14050 | None<br>African: 1/41950<br>None |
| 52 | 2098459417 | Ex43 | c.11217G>A | p.Trp3739Ter | ALFA | 0/10680 | None |
| 53 | 1230102996 | Ex43 | c.11222C>G | p.Ser3741Ter | GnomAD<br>ALFA | 2/139836<br>0/10680 | European: 2/75762<br>None |
| 54 | 2098459238 | Ex43 | c.11251G>T | p.Glu3751Ter | None | None | None |
| 55 | 1305040869 | Ex44 | c.11392C>T | p.Gln3798Ter | None | None | None |
| 56 | 2098446552 | Ex44 | c.11419C>T | p.Gln3807Ter | 8.3KJPN | 1/16760 | JAPANESE: 1/16760 |
| 57 | 1467862852 | Ex44 | c.11521C>T | p.Gln3841Ter | 8.3KJPN<br>ALFA | 1/16760<br>0/14050 | JAPANESE: 1/16760<br>None |
| 58 | 776397027 | Ex44 | c.11536C>T | p.Gln3846Ter | GnomAD_exome<br>ExAC | 1/205888<br>1/34048 | European: 1/107654<br>European: 1/18438 |

## Generation of *Dst-b*[E2610Ter] mice by CRISPR-Cas9 system

We attempted to induce G-to-T (Glu to Stop) point mutation in the *Dst-b*/*Bpag1b* using a previously described procedure (*Sato et al., 2018*). The sequence (5′-GCT ATC AGG AAA GAA CAC GG-3′) was selected as guide RNA (gRNA) target. The gRNA was synthesized and purified by GeneArt Precision gRNA Synthesis Kit (Thermo Fisher Scientific, MA) and dissolved in Opti-MEM (Thermo Fisher Scientific). In addition, we designed a 102-nt single-stranded DNA oligodeoxynucleotide (ssODN) donor for inducing c.8058 G>T of the *Dst-b1* (accession # NM_134448.4); the nucleotide T was placed between 5′- and 3′-homology arms derived from positions 8013–8057 and 8066–8114 of the *Dst-b1* coding sequence, respectively. This ssODN was ordered as Ultramer DNA oligos from Integrated DNA Technologies (IA, USA) and dissolved in Opti-MEM. The pregnant mare serum gonadotropin (five units) and the human chorionic gonadotropin (five units) were intraperitoneally injected into female C57BL/6J mice (Charles River Laboratories, Kanagawa, Japan) with a 48 hr interval, and unfertilized oocyte were collected from their oviducts. We then performed in vitro fertilization with these oocytes and sperm from male C57BL/6J mice (Charles River Laboratories, Kanagawa, Japan) according to standard protocols. 5 hr later, the gRNA (5 ng/µl), ssODN (100 ng/µl), and GeneArt Platinum Cas9 Nuclease (Thermo Fisher Scientific) (100 ng/µl) were electroplated to zygotes by using NEPA21 electroplater (Nepa Gene, Chiba, Japan) as previously reported (*Sato et al., 2018*). After electroporation, fertilized eggs that had developed to the two-cell stage were transferred into oviducts in pseudo-pregnant ICR female and newborns were obtained. To confirm the G-to-T point mutation induced by CRISPR/ Cas9, we amplified genomic region including target sites by PCR with the same primers used for PCR-RFLP genotyping. The PCR products were cut with *Xho*I for genome editing validation, and then, sequenced by using BigDye Terminator v3.1 Cycle Sequencing Kit (Thermo Fisher Scientific). We analyzed three independent *Dst-b* lines in this study: line numbers #1, # 6, and #7.

## Western blotting

Frozen heart, gastrocnemius muscle, and brain tissues were homogenized using a Teflon-glass homogenizer in ice-cold homogenization buffer (0.32 M sucrose, 5 mM EDTA, 10 mM Tris-HCl, and pH 7.4, phosphatase inhibitor cocktail tablet; Roche, Mannheim, Germany), centrifuged at 4500 rpm for 10 min at 4°C and the supernatants were collected. The protein concentration was determined using the bicinchoninic acid Protein Assay Reagent (Thermo Fisher Scientific). Lysates were mixed with an equal volume of 2× sodium dodecyl sulfate (SDS) sample buffer (125 mM Tris-HCl, pH 6.8, 4% SDS, 20% glycerol, and 0.002% bromophenol blue) for a final protein concentration of 2 µg/µl and denatured in the presence of 100 mM dithiothreitol at 100°C for 5 min. SDS-polyacrylamide gel electrophoresis (PAGE) was performed with 20 µg per lane on 5–20% gradient gels (197-15011; SuperSep Ace; FUJIFILM Wako Pure Chemical Corporation, Osaka, Japan) running at 10–20 mA for 150 min. The gels were blotted onto an Immobilon-P transfer membrane (Millipore, Billerica, MA). After blocking

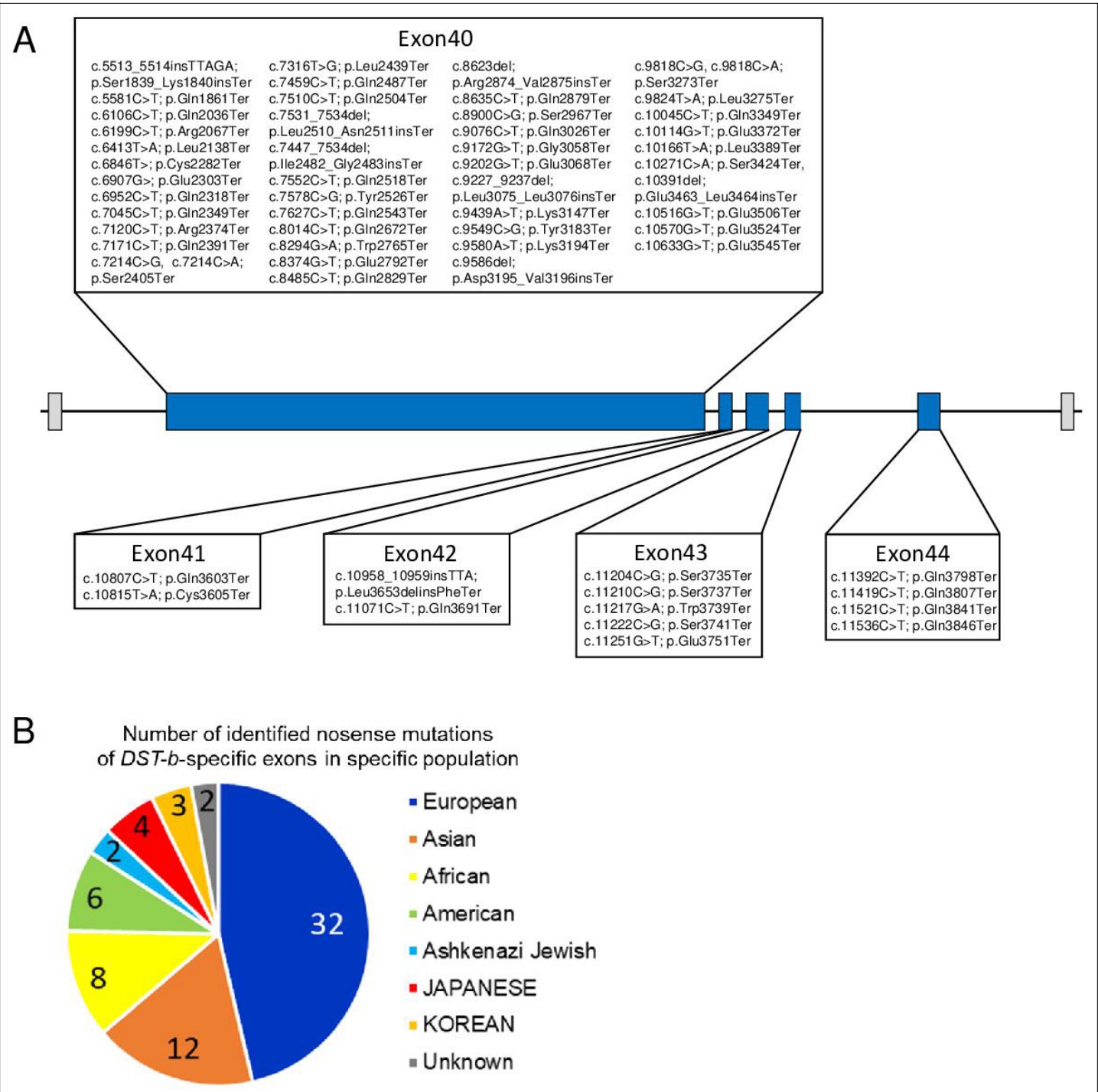

**Figure 10.** Nonsense mutations in the *DST-b*-specific exons. (**A**) Locations of nonsense mutations in the *DST-b*-specific exons on human chromosome 6 (NM_001374736.1). *DST-b*-specific exons are indicated blue rectangles. Nonsense mutations identified in the dbSNP database were distributed in all five *DST-b* specific exons. (**B**) Pie chart shows the frequency distribution of nonsense mutations of *DST-b*-specific exons in different populations. The colors indicate different populations. The numbers in the pie chart indicate the frequency of identified mutations in each population. *DST* variants identified in Japanese patients with myopathy are shown in *Supplementary file 1G*.

with 10% skim milk for 3 hr, blotted membranes were incubated with the following primary antibodies: rabbit polyclonal anti-Dst antibody (gifted from Dr. Ronald K Leim; 1:4000; *Goryunov et al., 2007*) that recognizes the plakin domain of Dst, and mouse monoclonal anti-glyceraldehyde-3-phosphate dehydrogenase (Gapdh) antibody (016-25523; clone 5A12, 1:10,000, Wako). Each primary antibody

was incubated overnight at 4°C. Then membranes were incubated with peroxidase-conjugated secondary antibodies for 1 hr at room temperature: anti-rabbit immunoglobulin G (IgG) (AB_2099233; Cat# 7074, 1:2000, Cell Signaling Technology), anti-mouse IgG (AB_330924; Cat# 7076, 1:2000, Cell Signaling Technology). Tris-buffered saline (10 mM Tris-HCl, pH 7.5, 150 mM NaCl) containing 0.1% Tween-20 and 10% skim milk was used for the dilution of primary and secondary antibodies, and Tris-buffered saline containing 0.1% Tween-20 was used as the washing buffer. Immunoreactions were visualized with ECL (GE Healthcare, Piscataway Township, NJ) or ImmunoStar LD (FUJIFILM Wako Pure Chemical) and a Chemiluminescent Western Blot Scanner (C-Digit, LI-COR, Lincoln, NE). The signal intensity of each band was quantified using Image Studio software version 5.2 (LI-COR).

## RNA extraction and real-time PCR

Total RNA was extracted from the heart, soleus, and brain using the RNeasy Mini Kit (QIAGEN, Hilden, Germany) including DNase digestion. 100 ng of RNA template was used for cDNA synthesis with oligo (dT) primers. Real-time PCR was performed using a StepOnePlus Real-Time PCR system (Thermo Fisher Scientific) and the following cycling conditions: 95°C for 2 min, followed by 40 cycles of 95°C for 15 s, 60°C for 40 s, and 95°C for 15 s. Gene expression levels were analyzed using the ΔΔCT method. *Actb* or *Gapdh* were used as internal controls to normalize the variability of expression levels. The primers used for real-time PCR are listed in *Table 2*.

## RNA-seq analysis

RNA-seq analysis was performed as described in the previous study with slight modification (*Bizen et al., 2022*; *Hayakawa-Yano et al., 2017*; *Hayakawa-Yano and Yano, 2019*). mRNA libraries were generated from total RNA 5 μg/sample extracted from the heart using illumine TruSeq protocols for poly-A selection, fragmentation, and adaptor ligation. Multiplexed libraries were sequenced as 150-nt paired-end runs on an NovaSeq6000 system. Sequence reads were mapped to the reference mouse genome (GRCm38/mm10) with Olego. Expression and alternative splicing events were quantified with Quantas tool. Integrative Genomics Viewer (IGV) was used as visualization of alignments in mouse genomic regions (*Thorvaldsdóttir et al., 2013*). Statics of differential expression was determined with edseR (*Robinson et al., 2010*). Differentially expressed genes were corrected with threshold set at p<0.01 and FDR < 0.1. GO analysis and KEGG pathway analysis were performed using DAVID Bioinformatics Resources (*Huang et al., 2009*) (https://david.ncifcrf.gov/summary.jsp). The threshold of GO analysis and pathway analysis was set at p<0.01. Three highly changed genes, *Xist*, *Tsix*, and *Bmp10,* were excluded because the differences were due to extreme outliers in only one individual. In GO analysis, the terms of cytoplasm and membrane that contained more than 190 genes were excluded from the list. PCA and hierarchical clustering were performed and visualized with R software using the RNA-seq datasets obtained from three independent WT and *Dst-b*$^{E2610Ter}$. PCA was conducted in the prcomp function in RStudio (version 2021.09.1 build 372). RNA-seq data can be accessed from the Gene Expression Omnibus under accession, GSE184101.

## Behavioral tests

The rotarod test and wire hang test (O'Hara & Co., Tokyo, Japan) were performed to evaluate motor coordination as described in the previous study with slight modification (*Horie et al., 2020*). In the rotarod test, the latency to fall from a rotating rod (30 mm diameter) with an acceleration from 10 to 150 rpm was measured. Each trial was conducted for a period of 3 min. In each mouse, two trials were conducted in a day. In the wire hang test, mouse was placed on the top of the wire lid. The lid was slightly shaken several times to force the mouse to grip the wires. The lid was turned upside down. The latency to fall was measured.

## Histological analysis

For tissue preparations, mice were euthanized via intraperitoneal injection with pentobarbital sodium (100 mg/kg body weight), and then perfused with 4% paraformaldehyde (PFA) in 0.1 M phosphate-buffered (PB) solution (pH 7.4). The tissues were fixed by cardiac perfusion with 0.01 M phosphate-buffered saline (PBS) followed by ice-cold 4% PFA in 0.1 M PB (pH 7.4). Dissected tissues were immersed in the same fixative overnight. To cut spinal cord and DRG sections, the specimens were rinsed with water for 10 min and decalcified in Morse solution (135-17071; Wako, Osaka, Japan)

**Table 2.** Primer list for qPCR.

| Gene name | Forward (5′ to 3′) | Reverse (5′ to 3′) |
| --- | --- | --- |
| Acta2 | GTCCCAGACATCAGGGAGTAA | TCGGATACTTCAGCGTCAGGA |
| Actb | GGCTGTATTCCCCTCCATCG | CCAGTTGGTAACAATGCCATGT |
| Ano5 | TCCAAAGAGACCAGCTTTCTCA | GTCGATCTGCCGGATTCCAT |
| Atp5a1 | TCTCCATGCCTCTAACACTCG | CCAGGTCAACAGACGTGTCAG |
| Cenpf | GCACAGCACAGTATGACCAGG | CTCTGCGTTCTGTCGGTGAC |
| Cox4i1 | ATTGGCAAGAGAGCCATTTCTAC | CACGCCGATCAGCGTAAGT |
| Ctgf | GGGCCTCTTCTGCGATTTC | ATCCAGGCAAGTGCATTGGTA |
| Dst-a | AACCCTCAGGAGAGTCGAAGGT | TGCCGTCTCCAATCACAAAG |
| Dst-b | ACCGGTTAGAGGCTCTCCTG | ATCACACAGCCCTTGGAGTTT |
| Dst isoform1 | TCCAGGCCTATGAGGATGTC | GGAGGGAGATCAAATTGTGC |
| Dst isoform2 | AATTTGCCCAAGCATGAGAG | CGTCCCTCAGATCCTCGTAG |
| Dst isoform3 | CACCGTCTTCAGCTCACAAA | AGTTTCCCATCTCTCCAGCA |
| Gapdh | AGGTCGGTGTGAACGGATTTG | TGTAGACCATGTAGTTGAGGTCA |
| Gpx1 | CCACCGTGTATGCCTTCTCC | AGAGAGACGCGACATTCTCAAT |
| Gpx4 | GCCTGGATAAGTACAGGGGTT | CATGCAGATCGACTAGCTGAG |
| Hmox1 | AAGCCGAGAATGCTGAGTTCA | GCCGTGTAGATATGGTACAAGGA |
| Hspa1l | TCACGGTGCCAGCCTATTTC | CGTGGGCTCATTGATTATTCTCA |
| Hspb1 | CGGTGCTTCACCCGGAAATA | AGGGGATAGGGAAAGAGGACA |
| Myh7 | ACTGTCAACACTAAGAGGGTCA | TTGGATGATTTGATCTTCCAGGG |
| Mylk4 | GGGCGTTTTGGTCAGGTACAT | ACGCTGATCTCGTTCTTCACA |
| Ndufb8 | TGTTGCCGGGGTCATATCCTA | AGCATCGGGTAGTCGCCATA |
| Nppa | GCTTCCAGGCCATATTGGAG | GGGGGCATGACCTCATCTT |
| Nppb | CATGGATCTCCTGAAGGTGC | CCTTCAAGAGCTGTCTCTGG |
| Nqo1 | AGCGTTCGGTATTACGATCC | AGTACAATCAGGGCTCTTCTCG |
| Rpl3l | GAAGGGCCGGGGTGTTAAAG | AGCTCTGTACGGTGGTGGTAA |
| Scn4a | AGTCCCTGGCAGCCATAGAA | CCCATAGATGAGTGGGAGGTT |
| Scn4b | TGGTCCTACAATAACAGCGAAAC | ACTCTCACCTTAGGGTCAGAC |
| Sdha | GGAACACTCCAAAAACAGACCT | CCACCACTGGGTATTGAGTAGAA |
| Sod1 | AACCAGTTGTGTTGTCAGGAC | CCACCATGTTTCTTAGAGTGAGG |
| Tgfb2 | TCGACATGGATCAGTTTATGCG | CCCTGGTACTGTAGATGGA |
| Tnni1 | ATGCCGGAAGTTGAGAGGAAA | TCCGAGAGGTAACGCACCTT |
| Tnni2 | AGAGTGTGATGCTCCAGATAGC | AGCAACGTCGATCTTCGCA |
| Tnnt1 | CCTGTGGTGCCTCCTTTGATT | TGCGGTCTTTTAGTGCAATGAG |
| Tnnt3 | GGAACGCCAGAACAGATTGG | TGGAGGACAGAGCCTTTTTCTT |
| Uchl1 | AGGGACAGGAAGTTAGCCCTA | AGCTTCTCCGTTTCAGACAGA |
| Uqcrc2 | AAAGTTGCCCCGAAGGTTAAA | GAGCATAGTTTTCCAGAGAAGCA |

**Table 3.** Primary antibodies for immunohistochemistry.

| Antigen name | Host | Dilution | Clone name | Source, Cat#, or reference |
|---|---|---|---|---|
| α-Actin | Mouse | 1:200 | Alpha Sr-1 | BioLegend, MMS-467S |
| αB-Crystallin | Rabbit | 1:1000 | | BioLegend, PRB-105P |
| ATF3 | Rabbit | 1:1000 | | Santa Cruz Biotechnology, sc-188 |
| Cytochrome C | Mouse | 1:500 | A-8 | Santa Cruz Biotechnology, sc-13156 |
| Desmin | Mouse | 1:100 | RD301 | Santa Cruz Biotechnology, sc-23879 |
| Desmin | Rabbit | 1:1000 | | Novus Biologicals, NBP1-85549 |
| Dst | Rabbit | 1:1000 | | Dr. Ronald K Liem, *Goryunov et al., 2007* |
| Lamin A/C | Mouse | 1:100 | E-1 | Santa Cruz Biotechnology, sc-376248 |
| LC3A/B | Rabbit | 1:1000 | D3U4C | Cell Signaling Technology, #12741 |
| Myotilin | Rabbit | 1:500 | | ProteinTech, 10731-1-AP |
| NF-M | Mouse | 1:200 | 1C8 | Dr. Katsuhiko Ono, *Horie et al., 2014* |
| p62 | Mouse | 1:200 | 1B5.H9 | BioLegend, MMS-5034 |
| p62 | Rabbit | 1:400 | | ABclonal, A19700 |
| Phospho-PERK (phospho T982) | Rabbit | 1:500 | | Abcam, ab192591 |
| Plectin | Mouse | 1:100 | 10F6 | Santa Cruz Biotechnology, sc-33649 |
| SUMO-1 | Mouse | 1:200 | D-11 | Santa Cruz Biotechnology, sc-5308 |
| SUMO-2/3 | Rabbit | 1:200 | | ABclonal, A5066 |
| Tom20 | Rabbit | 1:1000 | | Santa Cruz Biotechnology, sc-11415 |
| Ubiquitin | Rabbit | 1:1000 | | Dako; Agilent Technologies Z0458 |

overnight. Tissues were then dehydrated using an ascending series of ethanol and xylene washes, and then embedded in paraffin (P3683; Paraplast Plus; Sigma-Aldrich, St. Louis, MO). Consecutive 10-μm-thick paraffin sections were cut on a rotary microtome (HM325; Thermo Fisher Scientific), mounted on MAS-coated glass slides (Matsunami Glass, Osaka, Japan), and air-dried on a hot plate overnight at 37°C. Paraffin sections were deparaffinized in xylene, rehydrated using a descending series of ethanol washes, and then rinsed in distilled water. H&E staining and Masson's trichrome staining were performed by standard protocols.

For IHC, deparaffinized sections were treated with microwave irradiation in 10 mM citric acid buffer, pH 6.0 for 5 min, and incubated overnight at 4°C with the primaries listed in *Table 3*. All primary antibodies were diluted in 0.1 M PBS with 0.01% Triton X-100 (PBST) containing 0.5% skim milk. Sections were then incubated in horseradish peroxidase-conjugated secondary antibody (1:200; MBL, Nagoya, Japan) diluted in PBST containing 0.5% skim milk for 60 min at 37°C. Between each step, sections were rinsed in PBST for 15 min. After rinsing sections in distilled water, immunoreactivity was visualized in 50 mM Tris buffer (pH 7.4) containing 0.01% diaminobenzidine tetrahydrochloride and 0.01% hydrogen peroxide at 37°C for 5 min. Sections were then dehydrated through an ethanol-xylene solution and placed on coverslips with Bioleit (23-1002; Okenshoji, Tokyo, Japan). Digital images were taken with a microscope (BX53; Olympus, Tokyo, Japan) equipped with a digital camera (DP74, Olympus), and the TIF files were processed with Photoshop software (Adobe, San Jose, USA).

For immunofluorescent staining, sections were incubated in mixtures of Alexa488- or Alexa594-conjugated antibodies (1:200; Invitrogen, CA) for 60 min at 37°C. Cell nuclei were counterstained with DAPI (1:2000; Dojindo, Kumamoto, Japan) for 10 min at room temperature. Mounted sections were air-dried and coverslipped. Sections were observed and digital images were taken using a confocal laser scanning microscopy (FV-1200, Olympus). TIF files were processed with Adobe Photoshop software. Super-resolution images were recorded on the confocal laser scanning microscopy LSM 980 equipped with Airyscan 2 (Leica, Germany).

ISH was performed on paraffin sections as described in previous studies (*Takebayashi et al., 2000*) using following mouse probes: *PV,* also known as parvalbumin (*Pvalb*, GenBank accession, NM_013645, nt 92-885), *Nppa*, also known as atrial natriuretic peptide (*ANP*, GenBank accession, BC089615, nt 124-529), and *Nppb*, also known as brain natriuretic peptide (*BNP*, GenBank accession, D16497, nt 42-752, without intron sequence) were used.

## Electron microscope analysis

The detailed procedure for TEM analysis was described previously (*Shibata et al., 2015*). Briefly, the muscle samples for TEM were primary fixed with 2.5% glutaraldehyde for 24 hr at 4°C. Samples were washed in 50 mM HEPES buffer (pH 7.4) and were postfixed with 1.0% osmium tetroxide (TAAB Laboratories, England, UK) for 2 hr at 4°C. Samples were dehydrated with a series of increasing concentrations of ethanol (two times of 50, 70, 80, 90, 100% EtOH for 20 min each), soaked with acetone (Sigma-Aldrich) for 0.5 hr, and with n-butyl glycidyl ether (QY-1, Okenshoji Co. Ltd., Tokyo, Japan) two times for 0.5 hr, graded concentration of epoxy resin with QY-1 for 1 hr, and with 100% epoxy resin (100 g Epon was composed of 27.0 g MNA, 51.3 g EPOK-812, 21.9 g DDSA, and 1.1 ml DMP-30, all from Okenshoji Co. Ltd.) for 48 hr at 4°C, and were polymerized with pure epoxy resin for 72 hr at 60°C. Resin blocks with tissues were trimmed, semi-thin sliced at 350-nm-thickness stained with toluidine blue, and were ultrathin-sectioned at 80-nm thickness with ultramicrotome (UC7, Leica) by diamond knife (Ultra, DiATOME, Switzerland). The ultrathin sections were collected on the copper grids and were stained with uranyl acetate and lead citrate. The sections were imaged with TEM (JEM-1400 Plus, JEOL, Japan) at 100 keV.

## Measurements of ECG signal

Under anesthesia with 2–3% isoflurane (Pfizer Inc, NY), ECG signals were recorded. Three needle electrodes were inserted into right and left forelimbs as recording, and right hindlimb as grounding. The ECG signals were amplified using AC amplifier (band pass: 0.1–1 kHz), and the signals were digitized with A/D converter (Power 1401, Cambridge Electronic Design Ltd., Cambridge, UK).

## Quantification and statistical analysis

Morphometric analysis was performed at least three sections per mouse. Quantifications of cross-sectional area and fibrosis were performed with MetaMorph software (Meta Series Software version 7.10.2; Molecular Devices, San Jose, CA). Unless otherwise noted, sample size (n) is the number of animals in each genotype. For statistical analysis, Student's *t*-test and ANOVA were carried out. ANOVA was performed using ANOVA4 on the Web (https://www.hju.ac.jp/~kiriki/anova4/).

## In silico screening of *DST-b* mutations

In silico screening was performed using dbSNP database (*Sherry et al., 2001*) (https://www.ncbi.nlm.nih.gov/snp/?cmd=search). After downloading the results of a search using the keyword "DST" in dbSNP in TSV format, we extracted the entries whose "function_class" was "stop_gained".

## Compliance with ethical standards and study approval

All animal experiments were performed in accordance with the guidelines of the Ministry of Education, Culture, Sports, Science and Technology of Japan and were approved by the Institutional Animal Care and Use Committees of Niigata University (permit number: SA00521 and SA00621) and Tsukuba University (permit number: 17-078). Human study was approved by the ethical committees of the NCNP (permit number: A2019-123). The human materials used in this study were obtained for diagnostic purposes. The patients or their parents provided written informed consent for use of the samples for research.

# Acknowledgements

We thank Dr. Seiya Mizuno and Dr. Satoru Takahashi (Laboratory Animal Resource Center, Tsukuba University) for their contribution to generate *Dst-b* mutant mice, Dr. Kenji Sakimura and Dr. Manabu Abe for the provision of *Actb-iCre-IRES-GFP* mice, Dr. Kensuke Yamamura for electrophysiological apparatus, Dr. Ronald K H Liem for anti-Dst antibody, Dr. Keisuke Watanabe for *Pvalb* plasmid, and Dr. Riuko Ohashi and Mr. Kenji Oyachi (Histopathology Core Facility, Niigata University) for Masson's

trichrome staining. We also thank Dr. Norihisa Bizen for helpful discussion, Dr. Tomoko Shindo, Dr. Li Zhou, Mr. Yuya Imada, Ms. Satoko Yamagiwa, Ms. Yumi Kobayashi, Mr. Seiji Takahashi, Ms. Jitrapa Pinyomahakul, Ms. Aoba Shiina, and Mr. Osamu Arai for technical assistances. We thank Lisa Kreiner, PhD, from Edanz (https://www.jp.edanz.com/ac) for editing a draft of this manuscript. This work was supported by grants from JSPS (15H04667, 18H02592, 21H02652 to HT, 20K15912 to YN), The Uehara Memorial Foundation (HT), Nagai N-S Promotion Foundation for Science of Perception (HT), The Nakatomi Foundation (NY), Setsuro Fujii Memorial, Osaka Foundation for Promotion of Fundamental Medical Research (NY), LEGEND Research Grant from BioLegend (NY), and Niigata University Interdisciplinary Research Grant (NY).

## Additional information

### Competing interests

Ichizo Nishino: IN held grants from Sanofi, Daiicji Sankyo, Medical & Biological Laboratories, CYTOO, and Avidity Biosciences and holds a holds a patent on "Differentiating markers for inflammatory myopathies and methods to differentiate inflammatory myopathies" (Patent number 6531306). IN has received consultancy fees from Astellas, Sarepta, UCB Japan, Nobelpharma and Dyna Therapeutics, and has received honoria payments for lectures from Sanofi, Japan Blood Products Organization, Daiichi Sankyo, Mitsubishi Tanabe Pharma Corporation, Kyowa Kirin, Biogen, Asahi Kasei Pharma and Nihon Pharma. IN has participated either on the Advisory Board, Vice President or on the Executive board of Horizon Therapeutics, Astellas, World Muscle Society, Asian Oceanian Myology Center, Japan Muscle Society, Japanese Society of Neuropathology, Japanese Society of Neurotherapeutics, and Japanese Society of Neurology. The author has no other competing interests to declare. The other authors declare that no competing interests exist.

### Funding

| Funder | Grant reference number | Author |
|---|---|---|
| Japan Society for the Promotion of Science | 15H04667 | Hirohide Takebayashi |
| Japan Society for the Promotion of Science | 18H02592 | Hirohide Takebayashi |
| Japan Society for the Promotion of Science | 21H02652 | Hirohide Takebayashi |
| Japan Society for the Promotion of Science | 20K15912 | Nozomu Yoshioka |
| Uehara Memorial Foundation | Research grant | Hirohide Takebayashi |
| Nagai N-S Promotion Foundation for Science of Perception | Research grant | Hirohide Takebayashi |
| Nakatomi Foundation | Research grant | Nozomu Yoshioka |
| Setsuro Fujii Memorial, Osaka Foundation for Promotion of Fundamental Medical Research | Research grant | Nozomu Yoshioka |
| BioLegend | LEGEND Research Grant | Nozomu Yoshioka |
| Niigata University | Interdisciplinary Research Grant | Nozomu Yoshioka |

The funders had no role in study design, data collection and interpretation, or the decision to submit the work for publication.

## Author contributions
Nozomu Yoshioka, Conceptualization, Data curation, Formal analysis, Funding acquisition, Validation, Investigation, Visualization, Writing - original draft, Project administration; Masayuki Kurose, Shujiro Okuda, Yukiko Mori-Ochiai, Toshihiro Nagai, Ichizo Nishino, Shinsuke Shibata, Investigation; Masato Yano, Data curation, Software, Investigation, Writing - review and editing; Dang Minh Tran, Validation, Investigation; Masao Horie, Resources; Hirohide Takebayashi, Conceptualization, Resources, Supervision, Funding acquisition, Writing - original draft, Project administration

## Author ORCIDs
Nozomu Yoshioka http://orcid.org/0000-0002-4425-5457
Shujiro Okuda http://orcid.org/0000-0002-7704-8104
Hirohide Takebayashi http://orcid.org/0000-0003-4493-6604

## Ethics
Human study was approved by the ethical committees of the NCNP (Permit Number: A2019-123). The human materials used in this study were obtained for diagnostic purposes. The patients or their parents provided written informed consent for use of the samples for research.
All animal experiments were performed in accordance with the guidelines of the Ministry of Education, Culture, Sports, Science and Technology of Japan and were approved by the Institutional Animal Care and Use Committees of Niigata University (Permit Number: SA00521 and SA00621) and Tsukuba University (Permit Number: 17-078).

## Decision letter and Author response
Decision letter https://doi.org/10.7554/eLife.78419.sa1
Author response https://doi.org/10.7554/eLife.78419.sa2

# Additional files

## Supplementary files
• Supplementary file 1. (A) List of upregulated genes of Gene Ontology (GO) biological process, (B) list of upregulated genes of GO cellular component, (C) list of upregulated genes of Kyoto Encyclopedia of Genes and Genomics (KEGG) pathway, (D) list of downregulated genes of GO biological process, (E) list of downregulated genes of GO cellular component, (F) list of downregulated genes of KEGG pathway, and (G) variants of *DST* gene identified in Japanese patients with myopathy.

• Transparent reporting form

## Data availability
RNA-seq data can be accessed from the Gene Expression Omnibus under accession, GSE184101.

The following dataset was generated:

| Author(s) | Year | Dataset title | Dataset URL | Database and Identifier |
|---|---|---|---|---|
| Yoshioka N, Yano M, Takebayashi H | 2021 | mRNAseq in control(WT) and Dst-b mutant mouse heart | https://www.ncbi.nlm.nih.gov/geo/query/acc.cgi?acc=GSE184101 | NCBI Gene Expression Omnibus, GSE184101 |

The following previously published dataset was used:

| Author(s) | Year | Dataset title | Dataset URL | Database and Identifier |
|---|---|---|---|---|
| Hayakawa-Yano Y, Yano M | 2019 | mRNAseq in control (WT) and Nes:Cre_Qk cKO | https://metadataplus.biothings.io/geo/GSE123927 | NCBI Gene Expression Omnibus, GSE123927 |

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
