## [Editor Report]

The authors demonstrate that isoform-specific Dystonin-b (Dst-b) mutant mice show significant myopathy in skeletal and cardiac muscle at older ages without the peripheral neuropathy or postnatal lethality that are commonly observed by loss of function of the DST gene. The study provides novel information about the role of the Dst-b isoform in maintaining skeletal and cardiac muscle health. In addition, the study suggests that isoform-specific mutations in Dst-b gene may cause some hereditary skeletal and cardiac myopathies.

---

## [Decision Letter]

**Decision letter after peer review:**

Thank you for submitting your article "Isoform-specific mutation in Dystonin-b gene causes late-onset protein aggregate myopathy and cardiomyopathy" for consideration by *eLife*. Your article has been reviewed by 3 peer reviewers, and the evaluation has been overseen by a Reviewing Editor and Mone Zaidi as the Senior Editor. The following individual involved in the review of your submission has agreed to reveal their identity: Mark M Rich (Reviewer #1).

Essential revisions:

1. Figure 1E. If possible, please provide data on what percentage of mutant mice showed kyphosis. Please also provide individual hind limb muscle weight normalized with body weight.

2. Please quantify histologic outcomes shown in Figures1C, 1D and 4.

3. Please quantify the amount of fibrosis in Figures3C and 3D.

4. Please show data on desmin aggregates as supplemental material.

5. Please add quantification of mitochondrial content.

6. Please add independent verification of selected gene expression changes by qRT-PCR.

7. Please quantify myofiber CSA as recommended by Reviewer #3.

8. Please carefully consider the remaining comments from the reviewers.

*Reviewer #1 (Recommendations for the authors):*

Given that the majority of the paper is about muscle pathology, I was surprised that a few of the images of wild-type muscle cross-sections used for immunostaining were not of the quality I would expect. Two examples:

– In 3A the wild-type muscle has a great deal of space between fibers. That is an artifact. This is not a problem for interpretation per se, but it raises the question of whether that is the best wild-type section the authors have.

– In Figure 4B, the WT staining of actin does not show great striations. The staining of desmin in WT muscle in 4B versus 4F is very different. It is much nicer in 4F.

The discussion highlights the problem with the significance of the findings presented. It is a list of findings, but there are no big-picture discussion points. I believe this is because the significance of the findings and underlying mechanisms remain unknown.

*Reviewer #2 (Recommendations for the authors):*

A few new experiments and quantification of some of the histological data should improve the impact of the manuscript. Authors should also consider rigorously studying alternation in mitochondrial content and function in skeletal and cardiac muscle of mutant mice.

1) Histological alteration in Figures 1C, 1D, and Figure 4 should be quantified.

2) The amount of fibrosis in WT and mutant mice (Figure 3C, 3D) should be quantified.

3) What is the difference in Figure 4A top right and bottom left panel pictures?

4) Expression of a few mRNAs found to be altered in RNA-Seq experiment should be independently verified by performing QRT-PCR.

*Reviewer #3 (Recommendations for the authors):*

I would strongly recommend that authors perform a skeletal muscle regeneration study by cardiotoxin to see how muscle regeneration is affected by mutant Dst-b. This experiment can also highlight whether or not Dst-b mutant muscle stem cells have any defect in their ability and their contribution to the formation of new muscle fibers post injury.

---

## [Author Response]

Essential revisions:1. Figure 1E. If possible, please provide data on what percentage of mutant mice showed kyphosis. Please also provide individual hind limb muscle weight normalized with body weight.

Thank you for your suggestions. The kyphosis was observed in some (more than one third of) *Dst-b* mutant mice as shown in the author response image 1. MRI or CT imaging of the skeleton is necessary to accurately diagnose kyphosis, however, the imaging was not performed in this paper. Therefore, we would like not to provide data on what percentage of mutant mice showed kyphosis.

We weighed the soleus of hind limb and demonstrated the data (lines 132-135).

**Author response image 1. sa2fig1:** 

2. Please quantify histologic outcomes shown in Figures1C, 1D and 4.

As suggested, we quantified the histological data and demonstrated in Figures 2D-G and Figure 4B. Quantification data confirmed that neuropathy do not occur in *Dst-b^E2610Ter/E2610Ter^* mice and that desmin accumulates in the mutant mice.

3. Please quantify the amount of fibrosis in Figures3C and 3D.

According to the comment, we quantified the amount of fibrosis and demonstrate the data in Figure 3E.

4. Please show data on desmin aggregates as supplemental material.

We presented data on desmin aggregates in the cardiomyocytes of *Dst-b^E2610Ter/E2610Ter^* mice (Figure 4—figure supplement 1).

5. Please add quantification of mitochondrial content.

To address this issue, we quantified muscle fibers with mitochondrial accumulations (Figure 5B). Quantitative data confirmed that mitochondria accumulate in the myofibers of soleus in *Dst-b^E2610Ter/E2610Ter^* mice.

6. Please add independent verification of selected gene expression changes by qRT-PCR.

The reliability of RNA-seq was validated by real time-PCR of 13 genes (seven up-regulated genes and six down-regulated genes). The data was demonstrated in Figure 8—figure supplement 1C.

7. Please quantify myofiber CSA as recommended by Reviewer #3.

According to the comment, we quantified distribution of cross-sectional area (CSA) in the soleus. As shown in Figure 3C, small-caliber myofibers are abundant in *Dst-b^E2610Ter/E2610Ter^* mice.

8. Please carefully consider the remaining comments from the reviewers.

The remaining comments were also carefully considered and incorporated into this revised version as much as possible.

Reviewer #1 (Recommendations for the authors):Given that the majority of the paper is about muscle pathology, I was surprised that a few of the images of wild-type muscle cross-sections used for immunostaining were not of the quality I would expect. Two examples:– In 3A the wild-type muscle has a great deal of space between fibers. That is an artifact. This is not a problem for interpretation per se, but it raises the question of whether that is the best wild-type section the authors have.

Thank you very much for the comments. In this study, we used paraffin sections, which are versatile and usually retain tissue morphology in great detail. As reviewer#1 point out, space between myofibers is artifact which is pronounced in paraffin sections than fresh frozen sections. To address the concern, we present images of muscle fibers of fresh frozen sections from WT and *Dst-b^E2610Ter/E2610Ter^* mice as Author response image 2. We believe that the use of paraffin sections does not affect the conclusions, because a large number of centrally nucleated fibers (CNFs) were observed both in paraffin sections and fresh frozen sections.

– In Figure 4B, the WT staining of actin does not show great striations. The staining of desmin in WT muscle in 4B versus 4F is very different. It is much nicer in 4F.

Thank you very much for the pointing out. We replaced the images of Figure 4B with new image.

The discussion highlights the problem with the significance of the findings presented. It is a list of findings, but there are no big-picture discussion points. I believe this is because the significance of the findings and underlying mechanisms remain unknown.

Thank you very much for the critical comments. As pointed out, the mechanisms of *Dst-b* mutation-induced myopathy remain unknown. However, RNA-seq analysis provided molecular insights into pathophysiological mechanisms of cardiomyopathy in *Dst-b* mutant mice. For example, many genes responsible for unfolded protein response are affected (Figure 8C, *Hspa1l* and *Hspb1* in Figure 8—figure supplement 1C), which is similar to several animal models of myofibrillar myopathy (Winter et al., 2014; Fang et al., J Clin Invest, 2017). Moreover, we found nuclear inclusions in *Dst-b* mutant cardiomyocytes as a novel pathological hallmark. In the future, we would like investigate detailed molecular mechanisms underlying formation of protein aggregates.

The biological significance of this study would have been more obvious if we could have found MFM patients with DST-b mutations. However, we have not found them so far. We hope that this report will provide an opportunity to find MFM patients with DST-b mutations.

Reviewer #2 (Recommendations for the authors):A few new experiments and quantification of some of the histological data should improve the impact of the manuscript. Authors should also consider rigorously studying alternation in mitochondrial content and function in skeletal and cardiac muscle of mutant mice.1) Histological alteration in Figures 1C, 1D, and Figure 4 should be quantified.

As suggested, we quantified the histological data and demonstrated in Figures 2D-G and Figure 4B. Quantification data confirmed that neuropathy do not occur in *Dst-b^E2610Ter/E2610Ter^* mice and that desmin accumulates in the mutant mice.

2) The amount of fibrosis in WT and mutant mice (Figure 3C, 3D) should be quantified.

According to the comment, we quantified the amount of fibrosis and demonstrate the data in Figure 3E.

3) What is the difference in Figure 4A top right and bottom left panel pictures?

Thank you very much for the question. Figure 4A top right and bottom left are both soleus of *Dst-b* mutant mice. We would like to show desmin aggregation underneath the sarcolemma in the top right picture and desmin aggregates in the sarcoplasmic region in the bottom left picture.

4) Expression of a few mRNAs found to be altered in RNA-Seq experiment should be independently verified by performing QRT-PCR.

According to the comment, the reliability of RNA-seq was validated by real time-PCR of 13 genes (seven up-regulated genes and six down-regulated genes). The data was demonstrated in Figure 8—figure supplement 1C.

Reviewer #3 (Recommendations for the authors):I would strongly recommend that authors perform a skeletal muscle regeneration study by cardiotoxin to see how muscle regeneration is affected by mutant Dst-b. This experiment can also highlight whether or not Dst-b mutant muscle stem cells have any defect in their ability and their contribution to the formation of new muscle fibers post injury.

We agree that muscle regeneration study is very interesting subject. We would like to investigate on it in the future study.